# Astronomically calibrating early Ediacaran evolution

Tan Zhang [1,2,3], Chao Ma [1,2] ✉, Yifan Li [3] ✉, Chao Li [1], Anne-Christine Da Silva [4], Tailiang Fan[3], Qi Gao[5], Mingzhi Kuang[6], Wangwei Liu[7], Mingsong Li [8] & Mingcai Hou[1,2]

The current low-resolution chronostratigraphic framework for the early Ediacaran Period hampers a comprehensive understanding of potential trigger mechanisms for environmental upheavals and their connections to evolutionary innovation. Here, we establish a high-resolution astrochronological framework spanning ~57.6 million years of the early Ediacaran, anchored by the radioisotopic date of the Gaskiers glaciation onset, based on key sections from South China. Constrained by multiple radioisotopic dates, this framework precisely constrains the timing of the Marinoan deglaciation, Ediacaran Negative carbon isotope excursions 1 and 2 (EN1 and EN2), and key fossil assemblages (acanthomorphic acritarchs, Weng'an and Lantian biotas). These dates indicate the rapid termination of the Marinoan glaciation in South China within $10^6$-$10^7$ years, while providing robust temporal evidence for the global synchroneity of EN1, EN2, and Marinoan deglaciation. The integrated chronology refines the age model for early Ediacaran biotic evolution, revealing that ecosystems gradually increased in complexity over multi-million-year timescales while global taxonomic diversity remained relatively stable, punctuated by rapid transitions to novel communities coinciding with biogeochemical perturbations.

The Ediacaran Period (635–538.8 million years (Ma) ago) represents a pivotal epoch in Earth's history, characterized by profound transformations in the global climate, ocean chemistry, and biosphere[1]. The early Ediacaran (635–580 Ma ago) witnessed several key events: the Marinoan deglaciation, large carbon isotope excursions (CIEs) in marine carbonates (e.g., Ediacaran Negative excursions 1 and 2, EN1 and EN2)[2], and the emergence of new early eukaryotes and animals, evidenced by acanthomorphic acritarch assemblages and the Weng'an

and Lantian biotas[3]. A robust, high-resolution chronostratigraphic framework is crucial for deciphering the temporal details and causal mechanisms of these events. However, the current early Ediacaran chronostratigraphic framework suffers from limitations in resolution and remains subject to ongoing debates. It primarily relies on lithostratigraphic, chemostratigraphic, and biostratigraphic correlations, with only sparse radio-isotopic ages providing absolute dating constraints[4–6]. This chronological uncertainty poses significant

[1]State Key Laboratory of Oil and Gas Reservoir Geology and Exploitation & Institute of Sedimentary Geology, Chengdu University of Technology, Chengdu 610059, China. [2]Key Laboratory of Deep-time Geography and Environment Reconstruction and Applications of Ministry of Natural Resources, Chengdu University of Technology, Chengdu 610059, China. [3]School of Energy Resources, China University of Geosciences (Beijing), Beijing 100083, China. [4]Sedimentary Petrology Laboratory, University of Liege, Sart Tilman B20, Allée du Six Août 12, Liège 4000, Belgium. [5]College of Geography and Planning, Chengdu University of Technology, Chengdu 610059, China. [6]College of Energy, Chengdu University of Technology, Chengdu 610059, China. [7]Wuxi Research Institute of Petroleum Geology, Research Institute of Petroleum Exploration and Production, SINOPEC, Wuxi 214151, China. [8]Key Laboratory of Orogenic Belts and Crustal Evolution, MOE, School of Earth and Space Sciences, Peking University, Beijing 100871, China. ✉e-mail: machao@cdut.edu.cn; yifangeosci@gmail.com

challenges to advancing key research areas, limiting our understanding of this critical interval in Earth's history.

The snowball Earth hypothesis, which proposes globally synchronous and geologically rapid Cryogenian deglaciation events[1], faces challenges in verification due to significant chronological uncertainties. Although existing data from Australia, Namibia, and South China suggest a broadly simultaneous termination of the Marinoan glaciation, the lack of high-precision radiometric dates with extensive paleogeographic and stratigraphic coverage continues to obstruct a rigorous test of this synchronicity[4,7–9]. Current estimates for the depositional duration of the Marinoan cap dolostone vary substantially, ranging from tens of thousands of years to over 1.25 million years, depending on the models and techniques employed[4,6,10–13]. EN1 and EN2 events are interpreted as reflecting large-scale perturbations to the global dissolved inorganic carbon (DIC) reservoir, potentially associated with fluctuating marine redox conditions[2]. These CIEs are widely employed as key chronostratigraphic markers for regional to global stratigraphic correlation[3]. However, considerable debate surrounds their precise timing, reliability for reconstructing global carbon cycle dynamics and redox conditions, and their broader utility in stratigraphy. These debates largely stem from the temporal uncertainties inherent in current chronostratigraphic frameworks[2,12,14,15]. Moreover, the paucity of direct radiometric dates for early Ediacaran acritarch assemblages further complicates efforts to refine the temporal subdivision of the Ediacaran Period and reconstruct the evolutionary history of early animals[5,16]. The low resolution of the existing chronostratigraphic framework also exacerbates difficulties in testing hypotheses that link evolutionary innovations in the biosphere to large-scale biogeochemical perturbations, as uncertainties regarding both mechanistic links and chronological correlations persist.

Cyclostratigraphy refine the geological time scale by analyzing astronomically-forced climate cycles preserved in sedimentary records[17]. When tuned to an astronomical solution and integrated with radioisotope geochronology, it can produce a continuous, high-resolution astrochronology[17]. Cyclostratigraphy has been successfully applied to calibrate the Phanerozoic time scale, particularly in the Cenozoic and Mesozoic Eras[18]. The presence of well-preserved astronomical orbital signals in Mesoproterozoic and Paleoproterozoic sediments, some dating back as far as 2.48 billion years, highlights the potential for applying cyclostratigraphic methods to ancient sedimentary records (e.g., refs. 19,20). Previous cyclostratigraphic studies of the Ediacaran have primarily centered on the late Ediacaran Period (e.g., refs. 21,22), with relatively few investigations addressing the early Ediacaran (e.g., refs. 23,24). Current cyclostratigraphic efforts targeting the early Ediacaran face considerable challenges in South China. These challenges stem from uncertainties in establishing robust anchor points, which are complicated by two conflicting approaches for defining the top and thickness of the cap carbonate, and the absence of rigorous statistical evaluation of astronomical forcing[2,6,14,25,26]. Despite these challenges, the complete shallow-to-deep water sedimentary sequences in South China, coupled with abundant carbon isotopic data, fossil records, multiple radiometric dates and well-established stratigraphic frameworks, make it an exceptional region for studying both Ediacaran cyclostratigraphy and the co-evolution of life and environments (refs. 2–6,26–29). Notably, recent studies have identified the EN2 CIE in both South China and Newfoundland, precisely constrained to before 580.9 ± 0.4 Ma and immediately preceding the Gaskiers glaciation[30–32], making it a critical chronostratigraphic marker. The Gaskiers glaciation, a major climatic event documented across at least eight ancient continents and well-constrained by high-precision radiometric ages[33–35], presents potential for establishing a robust Ediacaran astronomical time framework.

Here, we establish a high-resolution astrochronological framework for the early Ediacaran Period by tuning astronomically-forced

cycles captured in magnetic susceptibility (MS) data from key successions in South China, which encompass both shallow platform to deep-water slope environments. This temporal framework, constrained by multiple radioisotopic dates and further supported by statistical methods evaluating sedimentation rates, provides precise constraints on the timing of Marinoan glaciation, key early Ediacaran CIEs and fossil records. Our integrated chronological framework, which synthesizes radiometric dates with global radioisotopic, chemostratigraphic, and redox data, offers a crucial chronometric context for testing hypotheses regarding the co-evolution of biological innovations and environmental changes during this pivotal interval in Earth's history.

## Results and discussion
### Geological setting and stratigraphy
At approximately 580 Ma, South China was positioned near paleo-(sub)tropical latitudes (15°N–35°N)[36] (Fig. 1a). The South China block was formed during the assembly of the supercontinent Rodinia through the amalgamation of the Yangtze and Cathaysia blocks[36] (Fig. 1b). The subsequent breakup of Rodinia initiated rifting, leading to the formation of rift basins that accumulated Neoproterozoic sediments[37]. The Ediacaran Doushantuo Formation, a siliciclastic-carbonate succession deposited on the Yangtze Block, displays a northwest-southeast facies transition from shallow-water platform to slope and basin environments[14,26] (Fig. 1b). The formation, with a thickness ranging from less than 40 m to over 100 m, consists predominantly of black shale and carbonate, and is traditionally subdivided into four lithostratigraphic members (Members I-IV) based on distinct lithological characteristics in the Yangtze Three Gorges area[26,27].

This study focuses on the lower to middle Doushantuo Formation, encompassing Members I-II and the lower part of Member III, as represented in three drillcores: (1) EYC2, from Quanshui Village, Yichang City, Hubei Province; (2) WD1, from Bainianguan Village, Wufeng County, Hubei Province; and (3) ZK68, from Cendongba Village, Songtao County, Guizhou Province (Fig. 1b). The studied intervals have thicknesses of 31.02 m, 115.4 m, and 95.5 m, respectively, and represent intrashelf basin (EYC2 and WD1) and lower slope (ZK68) environments (Fig. 1b). The EYC2 drillcore spans Member I and the lower part of Member II, while the WD1 drillcore covers the upper part of Member II and the lower part of Member III, creating a composite section that represents the majority of the lower-middle Doushantuo Formation in the intrashelf basin (Fig. 2a). Although a recent study proposed a potential stratigraphic hiatus near the boundary between Members II and III[38], our detailed core examination reveals no sedimentological evidence for such a discontinuity (Fig. 1c, d). The studied intervals of the Doushantuo Formation, which unconformably overlie the Cryogenian Nantuo Formation[27], primarily consist of dolostone, argillaceous dolomite, limestone, calcareous shale, and mudstone (Fig. 2b, o). The Doushantuo Formation has yielded an exceptional array of well-preserved fossil assemblages, including acanthomorphic acritarchs and the renowned Lantian, Weng'an, Wenghui, and Miaohe biotas[28,29,39]. These diverse fossil assemblages provide a critical biostratigraphic framework for defining and characterizing the Ediacaran System[3]. A comprehensive stratigraphic framework for the three studied drillcores has been established through the correlation of $\delta^{13}C_{carb}$ chemostratigraphy and lithostratigraphy from multiple localities across the Yangtze Platform, South China[26,27] (Fig. 2).

### $\delta^{13}C_{carb}$ and $^{87}Sr/^{86}Sr$ chemostratigraphy
The $\delta^{13}C_{carb}$ data from the studied intervals in the three drillcores reveal three negative and one positive CIEs: (1) EN1 near the Member I-II boundary, (2) WANCE (Weng'An Negative Carbon isotope Excursion)[40] in the middle of Member II, (3) EN2 near the Member II-III boundary, and (4) EP1 (Ediacaran Positive excursion 1) in the lower-

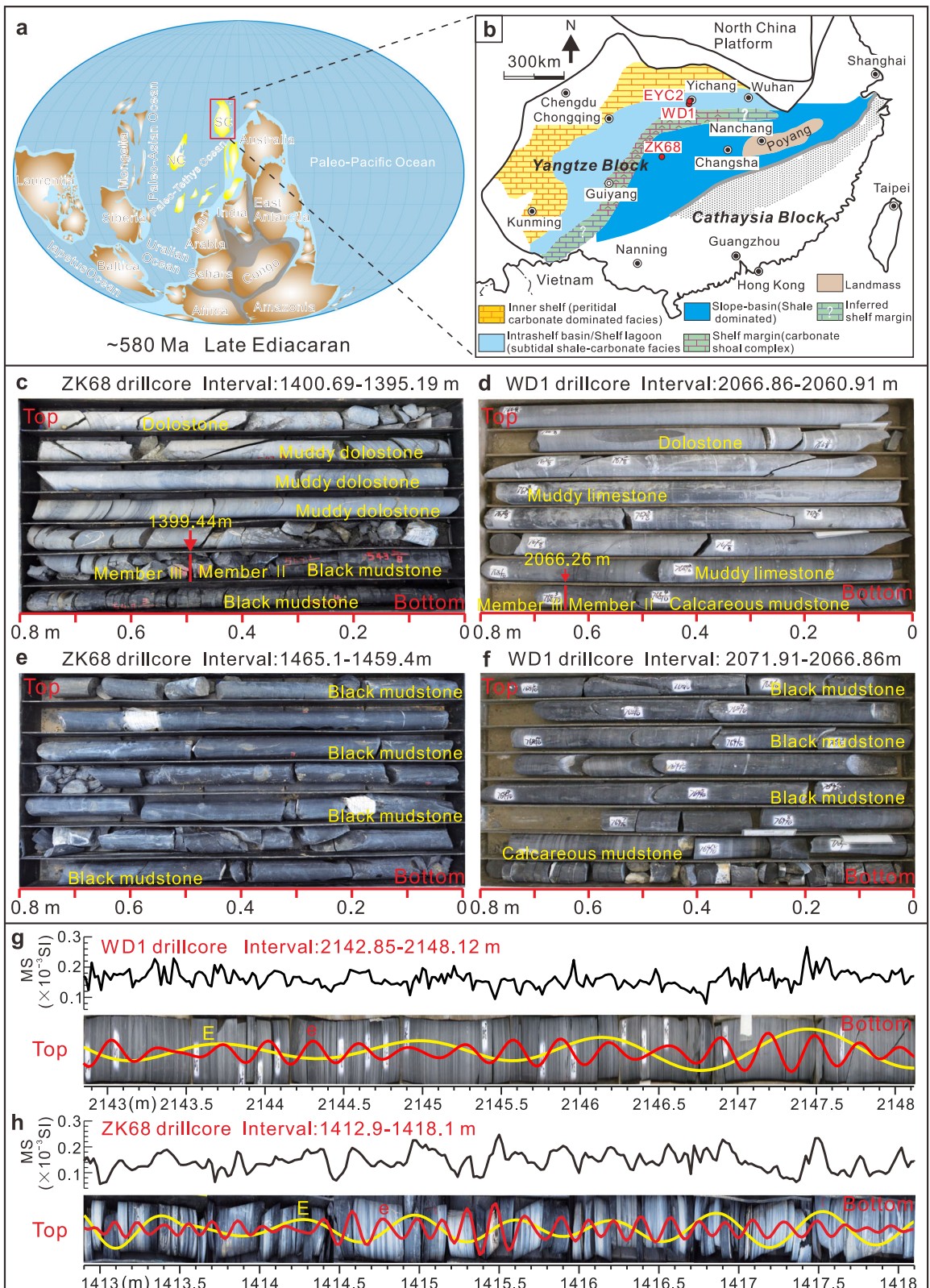

middle part of Member II (Fig. 2 and Supplementary Fig. 2). The measured $^{87}Sr/^{86}Sr$ ratios display three positive excursions (Fig. 2 and Supplementary Fig. 6) that coincide with the three negative $\delta^{13}C_{carb}$ excursions at approximately the same stratigraphic levels. Detailed descriptions of the evaluation of diagenetic alteration and chemostratigraphic correlation of the $\delta^{13}C_{carb}$ and $^{87}Sr/^{86}Sr$ records are provided in Supplementary Note 1.

## Cyclostratigraphic results

The multi-taper method (MTM) power spectrum analysis of the untuned MS series reveals distinct wavelengths throughout the entire stratigraphic intervals in the WD1, ZK68, and EYC2 drillcores (Supplementary Figs. 7–9). In the WD1 drillcore, dominant wavelengths are observed at 13.42 m, 4.97-1.34 m, 0.72-0.48 m, 0.19-0.15 m, and 0.11-0.084 m (Supplementary Fig. 7a). Similarly, the ZK68 drillcore displays

**Fig. 1 | Paleogeographic reconstruction of the Ediacaran Period and stratigraphic data. a** Global paleogeography at ~580 Ma[36]. **b** Paleogeographic reconstruction of the Ediacaran Yangtze Platform (modified from ref. 27), showing locations of studied drillcores (red circles). Reprinted from [Gondwana Research, 19, Ganqing Jiang, Xiaoying Shi, Shihong Zhang, Yue Wang, Shuhai Xiao, Stratigraphy and paleogeography of the Ediacaran Doushantuo Formation (ca. 635–551 Ma) in South China, 831–849, 2011], with permission from Elsevier. **c, d** Core photographs from the transitional interval between Members II and III in the ZK68 and WD1 drillcores, illustrating the lithological variations and continuous stratigraphic succession across this boundary. **e, f** Representative core

photographs showing black mudstone from the lower Member II of the ZK68 drillcore and the upper Member II of the WD1 drillcore. **g, h** Lamination cycles are closely reflected by variations in MS, as demonstrated by the 2142.85–2148.12 m interval in the WD1 drillcore and the 1412.9–1418.1 m interval in the ZK68 drillcore, which serve as two representative examples. The yellow and red curves represent the long eccentricity (**e**) and short eccentricity (**e**) cycles, respectively, extracted using a Gaussian filter with a bandpass of 0.46-0.96 cycles/m (**e**) and 2.78-3.96 cycles/m (**e**) for WD1, and 1.21-2.25 cycles/m (**e**) and 6.05–9.47 cycles/m (**e**) for ZK68. SC South China; NC North China; MS Magnetic susceptibility.

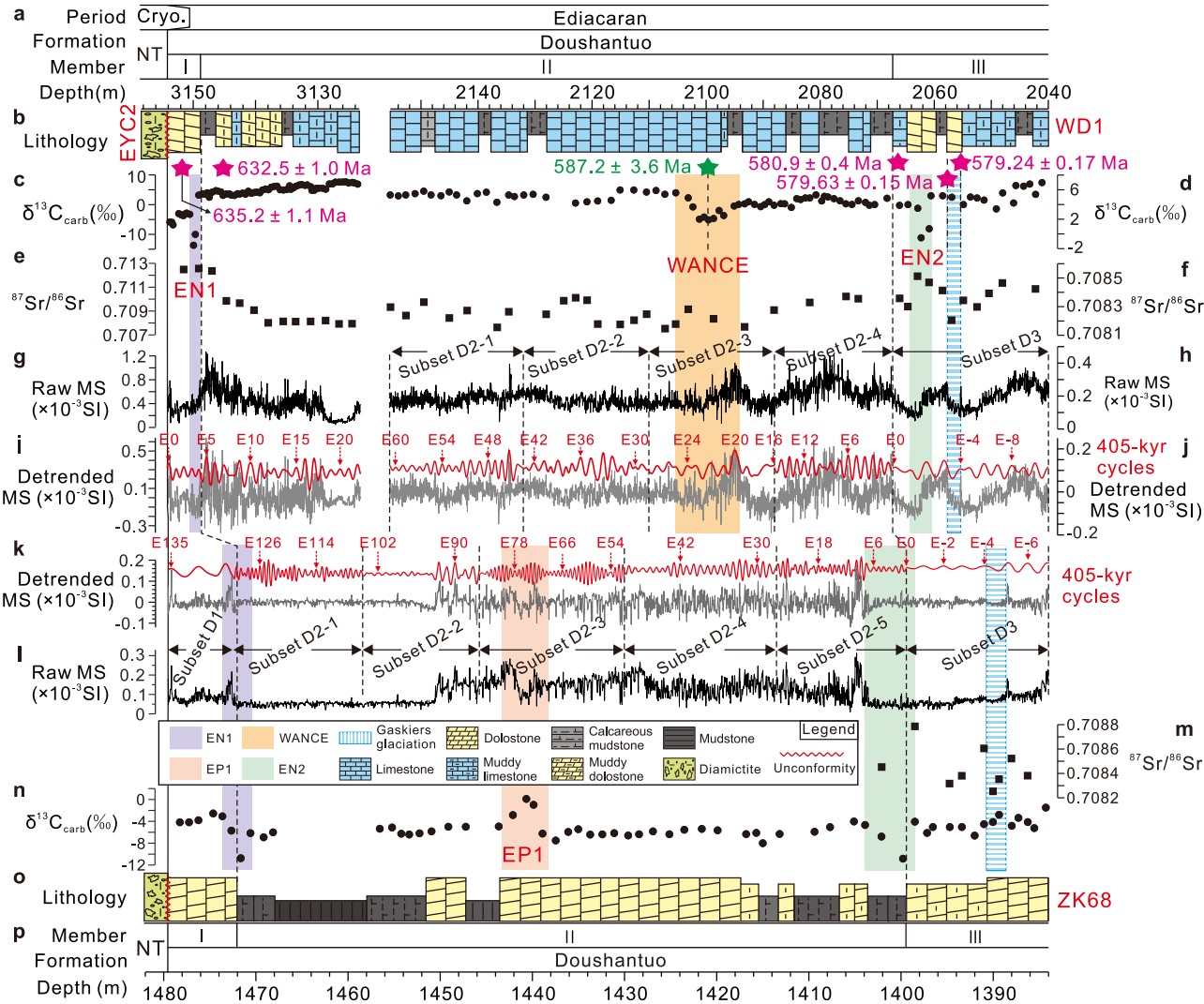

**Fig. 2 | Stratigraphic correlation between EYC2, WD1 and ZK68 drillcores and time series analyses. a–p** Chronostratigraphy and lithostratigraphy. **c–n** δ¹³C_carb data. **e–m** ⁸⁷Sr/⁸⁶Sr ratios. **g–l** Raw magnetic susceptibility (MS) data. **i–k** Extraction of ~405 kyr eccentricity cycles from MS data in EYC2, WD1, and ZK68 drillcores using Gaussian filters. EYC2: ~1.47 m cycles (0.675 ± 0.205 cycles/m). WD1: Five subsets: (D2-1) - 1.41 m cycles (0.71 ± 0.25 cycles/m), (D2-2) - 2 m cycles (0.50 ± 0.38 cycles/m), (D2-3) - 2.5 m cycles (0.40 ± 0.30 cycles/m), (D2-4) - 1.50 m cycles (0.665 ± 0.185 cycles/m), (D3) - 2.74 m cycles (0.365 ± 0.175 cycles/m). ZK68: Seven subsets−(D1) - 3.04 m cycles (0.329 ± 0.251 cycles/m), (D2-1) - 0.49 m cycles (2.05 ± 0.44 cycles/m), (D2-2) - 0.73 m cycles (1.37 ± 0.54 cycles/m), (D2-3) - 0.42 m

cycles (2.36 ± 0.28 cycles/m), (D2-4) - 0.68 m cycles (1.47 ± 0.29 cycles/m), (D2-5) - 0.57 m cycles (1.73 ± 0.52 cycles/m), (D3) - 2.10 m cycles (0.475 ± 0.155 cycles/m). The CA-ID-TIMS zircon U-Pb ages (in red) of 635.23 ± 0.57 Ma and 632.50 ± 0.48 Ma from ref. 6 and of 580.9 ± 0.4 Ma, 579.63 ± 0.15 Ma and 579.24 ± 0.17 Ma from refs. 30,35, and sediment Re-Os date (in green) of 587.2 ± 3.6 Ma from ref. 5 Details on the Gaskiers glaciation in South China are provided in Supplementary Note 4. MS, δ¹³C_carb, and ⁸⁷Sr/⁸⁶Sr data are in Supplementary Datas 1 and 2. Cryo. Cryogenian; NT. Nantuo Formation. EN1 Ediacaran Negative excursion 1; EN2 Ediacaran Negative excursion 2; EP1 Ediacaran Positive excursion 1; WANCE[40] Weng'An Negative Carbon isotope Excursion.

significant wavelengths at ~21.7 m, 5.62-0.50 m, 0.32-0.20 m, 0.08-0.065 m, and 0.05–0.041 m (Supplementary Fig. 8a). The EYC2 drillcore shows wavelengths of 3.52-1.25 m, 0.64-0.42 m, 0.16-0.13 m, and 0.09-0.07 m (Supplementary Fig. 9a). Notably, the ratios of these dominant wavelengths are largely consistent with those of the

theoretical Ediacaran orbital parameters[41,42] (see Supplementary Note 2 for details).

Evolutive Fast Fourier Transform (eFFT) analysis revealed frequency variations, which may reflect sedimentation rate changes. These variations, along with observed lithofacies changes and

lithostratigraphic boundaries, served as a framework for cyclostratigraphic analyses of multiple subsets, including five from WD1 and seven from ZK68 (Fig. 2 and Supplementary Figs. 7b, 8b and 9b). Integration of chemostratigraphic ($\delta^{13}C_{carb}$ and $^{87}Sr/^{86}Sr$ data), lithostratigraphic, and radioisotopic age constraints across the Yangtze Platform, South China, yields estimated durations and average sedimentation rates of: (1) - 0.79 Myr for Member I[4,6] and 0.32 cm/kyr in EYC2 and 0.95 cm/kyr in ZK68; (2) - 55.2 Myr for Member II in ZK68[6,31,35] and 0.13 cm/kyr; (3) - 7.2 Myr from the WANCE nadir to Member II top in WD1[5,35] and 0.45 cm/kyr; and (4) - 400 kyr for the Gaskiers glaciation interval in the lower Member III[35] and 0.56 cm/kyr in WD1 and 0.40 cm/kyr in ZK68 (see the "Estimation of average sedimentation rate" section for details). TimeOpt statistical[43,44] analysis reveals distinct sedimentation rates (cm/kyr) across the studied sections: 0.35 for Member II in EYC2; 0.35-0.37 and 0.64 for Members II and III in WD1; and 0.66, 0.10–0.18, and 0.53 for Members I, II, and III in ZK68 (see the "Time series methods" section and Supplementary Text 1 for details; Supplementary Figs. 10–24). Sedimentation rates derived from independent chronological constraints are consistent with results revealed by the TimeOpt method.

Cantine et al. [45] suggest that the low net sedimentation rates along the eastern Gondwanan margin during the early Ediacaran cannot be fully explained by the Sadler effect, which reflects the apparent decline in sedimentation rates over time due to non-steady accumulation. Instead, the low rates in Member II are attributed to environmental and depositional factors unique to the early Ediacaran. Rapid sea-level rise following the Marinoan deglaciation established a highstand systems tract on the Yangtze Platform[1,2,27], gradually submerging paleohighs and reducing terrigenous input due to the South China Block's mid-latitude position (ca. 35–45°N)[36] during ca. 635–580 Ma. Limited sediment supply and reduced carbonate productivity in the intrashelf basin and slope environments, consistent with the outer non-skeletal carbonate zone (paleolatitude ≥30°), likely contributed to low sedimentation rates[46]. Radiometric dating at the Jiulongwan section, a key intrashelf basin site on the Yangtze Platform, confirms similarly low sedimentation rates in the lower Member II (~0.23 cm/kyr)[24]. Average sedimentation rates decrease basinward, from ~0.4 cm/kyr at WD1 (shallow intrashelf basin) to ~0.34 cm/kyr at EYC2 (deep intrashelf basin) and ~0.17 cm/kyr at ZK68 (lower slope) (Supplementary Data 2), aligning with the depositional environments and paleogeography of the Yangtze Platform. Although detailed core examinations revealed no evidence of small-scale stratigraphic hiatuses or condensed intervals, detecting such features in core records remains inherently challenging[47]. Despite these uncertainties, the sedimentation rates align with independent radiometric dating and support a robust interpretation consistent with the geological context.

Based on the estimated optimal mean sedimentation rates, the 6.16–1.34 m, 6.26–0.44 m, and 1.82-1.25 m cycles in the WD1, ZK68, and EYC2 drillcores, respectively, are interpreted to correspond to the ~405 kyr eccentricity cycles (Supplementary Figs. 7–9). In the EYC2 drillcore, the amplitude modulation of the presumed long eccentricity cycle and the ~18.5 kyr precession signal produces envelopes with periods of ~2.0 Myr and ~103 kyr, respectively, closely matching the ~2.4 Myr eccentricity modulation and ~100 kyr precession modulation (Supplementary Fig. 25). Similarly, in the WD1 and ZK68 drillcores, the amplitude of the interpreted short eccentricity cycle is modulated in bundles of ~3.5–4 cycles, consistent with long eccentricity modulating short eccentricity (Supplementary Figs. 25–26). Spectral analysis of the Hilbert transform of the presumed precession, short eccentricity, and long eccentricity signals further identifies distinct power within the short eccentricity, long eccentricity, and ~2.4 Myr eccentricity modulation bands (Supplementary Figs. 25 and 27). These findings confirm that the MS data from the EYC2, WD1, and ZK68 drillcores meet a critical diagnostic test for orbital forcing, as both short and long

eccentricity cycles are successfully extracted from precession and short eccentricity signals[19,48].

These sedimentary cycles were isolated using Gaussian bandpass filtering (Fig. 2) and subsequently tuned to the long eccentricity cycles, resulting in floating astronomical time scales (ATS) of 9.12 Myr, 28.88 Myr, and 57.59 Myr for the EYC2, WD1, and ZK68 drillcores, respectively (Fig. 3 and Supplementary Data 2). The MTM power spectra of the ~405-kyr-tuned MS series in the three drillcores exhibit significant spectral peaks at ~405 kyr, 133–97.7 kyr, 33.3–25 kyr and 18.6–15.3 kyr (Supplementary Fig. 28), which align with major periods predicted for orbital forcing of early Ediacaran solar insolation[41,42]. The spectra also reveal longer-period cycles of 8.5 Myr, 5.5 Myr,, 2.8-1.9 Myr and 1.2 Myr, which may correspond to the modulation periods of the ~9 Myr, ~4.5 Myr and ~2.4 Myr orbital eccentricity cycles, as well as-1.2 Myr orbital obliquity cycles documented in Phanerozoic sedimentary records, respectively (e.g., refs. 49–51). More details on the cyclostratigraphic interpretation are provided in Supplementary Note 3.

## Constructing and testing a radioisotopically anchored early Ediacaran ATS

The 579.63 ± 0.15 Ma age (CA-ID-TIMS date) from the lower Rocky Harbour Formation on Bonavista Peninsula, Newfoundland, provides a precise constraint for the synglacial onset of the Gaskiers glaciation[35]. Due to its high precision and minimal uncertainty in marking the onset of the Gaskiers glaciation, we adopt the 579.63 ± 0.15 Ma age as the anchor point for constructing the Ediacaran ATS (see Supplementary Note 4 and Supplementary Fig. 30 for more details on the Gaskiers glaciation in South China). Anchoring long eccentricity calibrated MS series from the ZK68 and WD1 drillcores to 579.63 ± 0.15 Ma yields two anchored ATSs (Fig. 3): (1) 636.05 ± 0.43 to 578.42 ± 0.43 Ma for the ZK68 drillcore (lower slope), and (2) 605.25 ± 0.26 to 576.40 ± 0.26 Ma for the WD1 drillcore (intrashelf basin). For the EYC2 drillcore, also situated in an intrashelf basin setting, we construct an anchored ATS spanning from 635.73 ± 0.45 Ma to 626.61 ± 0.45 Ma (Fig. 3). This was achieved by provisionally anchoring the floating ATS to 634.90 ± 0.43 Ma, an age corresponding to the top of Member I, as suggested by the ZK68 drillcore (Fig. 3).

The ATS carries the following uncertainties: (1) an error of ± 0.15 Myr in the 579.63 ± 0.15 Ma U-Pb date marking the Gaskiers glaciation onset in WD1 and ZK68 drillcores (Anchor point 1); (2) an error of ±0.43 Myr in the 634.90 ± 0.43 Ma age of the top of Member I (Anchor point 2); (3) uncertainties in precisely determining the position of Gaskiers glaciation onset in the studied drillcores based on $^{87}Sr/^{86}Sr$ and MS data lead to errors of ± 0.04 Myr for WD1 and ± 0.025 Myr for ZK68 (see Supplementary Note 4 for more details), and (4) the uncertainty of spectral peak assignments in the cyclostratigraphic signal due to nonlinear climatic responses, which could have caused variable time lags between orbital forcing and sedimentation cyclic expression; here, we follow the previously proposed assumption of ±0.10[24]. Recently, Zeebe and Lantink[41,42], using advanced solar system integrations, suggest that the assumption of presumed 405 kyr eccentricity "metronome" becomes unreliable beyond ~500 Ma. This is attributed to potential instability in its period and a weakened long-eccentricity amplitude caused by the secular resonance $\sigma_{12} = (g_1 - g_2) + (s_1 - s_2)$ (see refs. 41,42,52 for further details). Based on sedimentation rates constrained by independent chronological data and the statistical TimeOpt method, the observed cycles in this study and their ratios are consistent with the theoretical Milankovitch cycles derived from the astronomical solution ZB23-N64[41,42]. Furthermore, the MS data from the EYC2, WD1, and ZK68 drillcores reveal a clear signal of long eccentricity modulation, as evidence by eccentricity-modulated precession and long eccentricity modulating short eccentricity (Supplementary Figs. 25–27). Beyond 500 Ma, the instability of the long-eccentricity cycle introduces uncertainty in

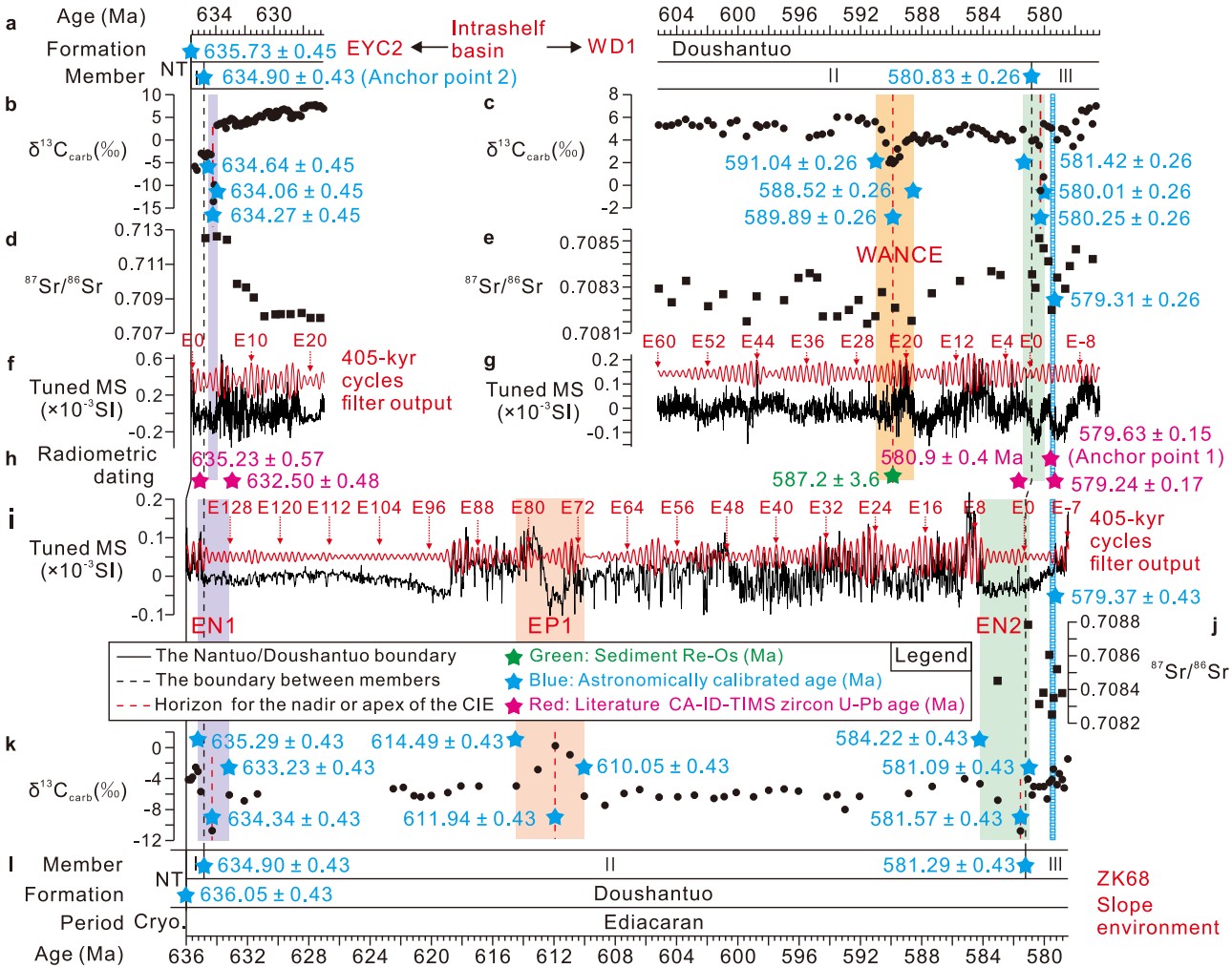

**Fig. 3 | The radioisotopically anchored astronomical time scales for the early Ediacaran Period. a, l** Astrochronological framework for the lower-middle Doushantuo Formation. **b, c, k** Tuned δ¹³Ccarb data. **d–j** Tuned ⁸⁷Sr/⁸⁶Sr ratios. **f–i** Tuned MS series. The ~405 kyr long eccentricity cycles (red curve) were extracted using a Gaussian filter with all passbands: 0.00247 ± 0.00050 cycles/kyr. The "E" in the ZK68 and WD1 drillcores is numbered sequentially from the Member II/III boundary downward, whereas in the EYC2 drillcore, the "E" is numbered from the base of the Doushantuo Formation upward. Age models for the WD1 and ZK68 drillcores are anchored to the CA-ID-TIMS date of 579.63 ± 0.15 Ma (Anchor point 1), marking the onset of Gaskiers glaciation[35], while that for the EYC2 drillcore is anchored to the astronomically calibrated age of 634.90 ± 0.43 Ma (Anchor point 2) for the Member I/II boundary from the ZK68 drillcore. **h** The age models are supported by published radiometric dating: 635.23 ± 0.57 Ma[6], 632.50 ± 0.48 Ma[6], 587.2 ± 3.6 Ma[5], 580.9 ± 0.4 Ma[30,35], and 579.24 ± 0.17 Ma[35]. Detailed information regarding age models and floating ATS is provided in Supplementary Data 2. EN1 Ediacaran Negative excursion 1; EN2 Ediacaran Negative excursion 2; EP1 Ediacaran Positive excursion 1; WANCE[40] Weng'An Negative Carbon isotope Excursion.

constructing the ATS. The ZB23-N64 solution predicts that at 610 Ma (the approximate mean age of the studied Doushantuo Formation in this study), the long eccentricity cycle predominantly exhibits a period of 405.28 ± 2.76 kyr[41,42,52] (Supplementary Data 3). Therefore, we adopt the long-eccentricity period of 405.28 kyr as the tuning target, with ±2.76 kyr representing the uncertainty associated with each long-eccentricity cycle. The cumulative nature of this uncertainty over successive cycles is identified as a fifth major source of error in the ATS. This cumulative effect leads to calculated errors of 0.39 Myr for the ZK68 drillcore (143 cycles × 2.76 kyr per cycle), 0.19 Myr for the WD1 drillcore (71 cycles × 2.76 kyr per cycle), and 0.063 Myr for the EYC2 drillcore (23 cycles × 2.76 kyr per cycle). Here, total uncertainties are calculated by summing all error sources in quadrature, taking the square root of the sum of the squares of individual uncertainties, ensuring a comprehensive and accurate assessment of the combined uncertainty[53–55]. Specifically, the overall uncertainties are estimated to be ± 0.43 Myr for ZK68 (calculated from $\sqrt{0.15^2 + 0.025^2 + 0.1^2 + 0.39^2}$), ± 0.26 Myr for WD1 (calculated from $\sqrt{0.15^2 + 0.04^2 + 0.1^2 + 0.19^2}$) and ± 0.45 Myr for EYC2 (calculated from $\sqrt{0.43^2 + 0.1^2 + 0.063^2}$).

The radioisotopically anchored ATSs (Fig. 3) are further constrained by four additional radioisotopic dates. First, two CA-ID-TIMS U-Pb zircon ages from South China corroborate the calculated age of 634.90 ± 0.43 Ma for the Member I/II boundary of the Doushantuo Formation: (1) 635.23 ± 0.57 Ma[6] from an ash bed 1.3 m below the top of Member I at Wuhe-Gaojiaxi section, and (2) 632.5 ± 0.48 Ma[6] from an ash bed ~5 m above the top of Member I at Jijiawan section. Second, Re-Os dating of a horizon 58 m above the base of the Doushantuo Formation in the Jiulongwan section yielded an age of 587.2 ± 3.6 Ma[5], correlating with both the WANCE event nadir and Sequence 1/2 boundary[5,26] (Supplementary Fig. 31, further discussions of these correlations are provided in Supplementary Note 5), which supports the 589.89 ± 0.26 Ma estimate for the equivalent horizon in the WD1 drillcore (Fig. 3). Third, the termination of the Gaskiers glaciation, dated at 579.24 ± 0.17 Ma on the Bonavista Peninsula using CA-ID-TIMS U-Pb methods[35], aligns well with our estimated ages of

579.37 ± 0.43 Ma (ZK68) and 579.31 ± 0.26 Ma (WD1) for the end of the associated cooling period in South China (Fig. 3). These agreements between radioisotopic dates and our astronomically tuned time scale confirm the reliability of this chronology and demonstrate the viability of extending ATS construction throughout the early Ediacaran Period.

The base of Member III of the Doushantuo Formation in South China is astronomically calibrated to 581.29 ± 0.43 Ma in the slope setting (ZK68) and 580.83 ± 0.26 Ma in the intrashelf basin (WD1) (Fig. 3). These dates reveal synchronous boundaries between individual members within the Doushantuo Formation across both platform and slope settings. Our astrochronological model clarifies the chronological implications of boundaries across different facies in the Ediacaran lithostratigraphic sequences of South China, refining the regional lithostratigraphic correlation framework.

## High-resolution chronology of Marinoan deglaciation

Our ATSs provide high-precision constraints on the Marinoan deglaciation in South China, a pivotal event characterized by an abrupt transition from Cryogenian glacial diamictite to Ediacaran postglacial cap dolostone at the base of the Doushantuo Formation. Specifically, our data constrain the onset of Marinoan deglaciation between 636.05 ± 0.43 Ma and 635.73 ± 0.45 Ma (Fig. 3) in South China. The deglaciation termination, marked by the cessation of cap dolostone deposition (Members I/II boundary), is precisely dated at 634.90 ± 0.43 Ma (Fig. 3). These constraints align well with previously reported Marinoan deglaciation ages, including: <632.3 ± 5.9 Ma (Re-Os date) from NW Canada[56], <635.21 ± 0.59 Ma (CA-TIMS date) from Namibia[8], <636.41 ± 0.45 Ma (CA-TIMS date) from Australia[7], and 636.8 ± 0.7 to 635.2 ± 0.6 Ma (astrochronology) from South China[24]. Previous U-Pb zircon ages from South China, including 634.57 ± 0.88 Ma from the uppermost Nantuo Formation[4] and 635.23 ± 0.57 Ma[6] from 1.3 m below the top of the Ediacaran cap dolostone likely overestimate the actual timing of cap dolostone deposition. This overestimation is attributed to potential stratigraphic hiatuses associated with the former age and the fact that the latter age predates the deposition of the uppermost cap dolostone. Our dates address these issues, offering tighter constraints on the onset of the Marinoan deglaciation and subsequent cap dolostone deposition in South China.

Estimates of cap carbonate deposition timescales, derived from various methods such as sedimentary structure interpretation, paleomagnetic reversal records, and radiometric dating−have produced widely varying results, ranging from $10^3 - 10^7$ years[4,6,10–13,57,58] (Table 1). Our astrochronological analysis indicates that cap dolostone deposition in South China occurred over a timescale of $10^6$ to $10^7$ years, which falls within the upper range of previously reported estimates. The recently proposed three-stage formation model of "Seafloor weathering-Continental weathering-Ocean Mixing (SCOM)"[59] provides a mechanism for reconciling diverse cap carbonate deposition timescales. This model integrates glacial seafloor weathering, post-glacial continental weathering, and meltwater-deep sea mixing processes. It posits rapid initial deposition in a stratified ocean ($10^4$-$10^5$ years), followed by slower deposition in a mixed ocean (up to ~5 million years), and concluding diagenetic processes[59]. This approach effectively synthesizes timescales from sedimentology, paleomagnetism, radiometric dating, and astrochronology, offering a coherent explanation for the observed complexities in cap carbonate deposition. The rapid Marinoan deglaciation in South China, synchronous with global Cryogenian events, signifies a dramatic shift from a frozen world to a hot, high-$CO_2$ environment[12], offering crucial insights into early Earth climate dynamics during this critical period.

## Chronology and global synchronicity of early Ediacaran CIEs

Examinations of Ediacaran CIEs within developed ATSs and existing geochronological data reveals intricate temporal details and global synchroneity during the 636−578 Ma interval. During this period, three significant CIEs, namely EN1, WANCE, and EN2, were recorded across platformal to slope-basinal settings in South China, representing some of the best-characterized oxygenation events in the Ediacaran ocean[2]. While EN1 and EN2 exhibit global distribution[3], the WANCE event appears primarily confined to South China[40,60].

The EN1 event, globally preserved in the basal Ediacaran cap dolostones, exhibits variable durations across diverse depositional settings in South China. In the intrashelf basin, EN1 spans 0.58 Myr (634.64 ± 0.45 to 634.06 ± 0.45 Ma, with a nadir at 634.27 ± 0.45 Ma), while in the lower slope environment, it extends over 2.06 Myr (635.29 ± 0.43 to 633.23 ± 0.43 Ma, nadir at 634.34 ± 0.43 Ma) (Fig. 3). These ages are consistent with previously reported age estimates of ~636−632 Ma for the basal Ediacaran cap dolostone worldwide[4,6−8,56], corroborating the global synchroneity of EN1. The WANCE event, occurring between EN1 and EN2 in the Yangtze Gorges area of South China, represents a regional oxidation event[60], astronomically constrained between 591.04 ± 0.26 Ma and 588.52 ± 0.26 Ma, with a nadir of 1.9 ‰ at 589.89 ± 0.26 Ma (Fig. 3). This oxygenation event signifies the pulsed oxidation of a substantial marine dissolved organic carbon (DOC) reservoir, facilitating intermittent expansion of oxygenated benthic zones and potentially catalyzing the evolution of more intricate ecosystems[60].

EN2 exhibits global correlation, corresponding to CIEs in the upper Masirah Bay Formation of Oman[61], the upper Karibib Formation of northern Namibia[62], and the upper Mall Bay Formation in Newfoundland, Canada[30]. On the Avalon Peninsula, Newfoundland, the CIE in the upper Mall Bay Formation is located several tens of meters below the base of the Gaskiers Formation. CA-ID-TIMS dating of an ash bed 7.75 m beneath the base of the Gaskiers Formation suggests that this CIE is slightly older than 580.90 ± 0.40 Ma[32,35]. On the Yangtze Platform in South China, the duration of EN2 varies with depositional setting. In the intrashelf basin (WD1), EN2 lasted 1.41 Myr, from 581.42 ± 0.26 to 580.01 ± 0.26 Ma, nadir at 580.25 ± 0.26 Ma. In the lower slope (ZK68), it persisted for 3.13 Myr, from 584.22 ± 0.43 to 581.09 ± 0.43 Ma, with a nadir at 581.57 ± 0.43 Ma (Fig. 3). These age constraints from South China are consistent with the estimated age for EN2 in Newfoundland, supporting the global synchroneity of this event.

The early Ediacaran CIEs (EN1 and EN2) exhibit significant spatiotemporal heterogeneity across diverse depositional environments, evident in variations of onset, duration, and rates of isotopic change (Table 2). Significantly, the isotope change rates exhibit a distinctive dynamic pattern: a notably slower rate of isotopic decline during the onset phase compared to the rapid isotopic recovery (Table 2). EN1, interpreted as evidence of gas-hydrate destabilization during postglacial warming[14], shows a gradual onset reflecting progressive hydrate destabilization, followed by accelerated methane release and a rapid recovery due to light carbon depletion and negative feedback mechanisms (e.g., enhanced biological pump activity and accelerated silicate weathering). EN2, attributed to enhanced oxidation of a large DOC pool in the Ediacaran ocean, is closely associated with elevated weathering-derived oxidants and nutrients[2]. The invariance of the carbon isotope composition of organic carbon ($\delta^{13}C_{org}$) across EN2 implies that the DOC pool size remained largely unchanged[2]. In the context of the predominantly anoxic ocean conditions[63,64], the slow onset of EN2 likely reflects the progressive increase in oxidant availability (e.g., sulfate and/or evaporite)[65,66]. Conversely, its rapid recovery may result from diminished terrestrial sulfate input and swift reestablishment of ocean anoxia. This pattern is substantiated by the strontium isotope ($^{87}Sr/^{86}Sr$) trend during EN2, which shows an inverse correlation with the $\delta^{13}C_{carb}$ curve (Figs. 2 and 4, and Supplementary Fig. 6).

The observed variability and asymmetric pattern, while indicative of global phenomena, suggest that the expression of EN1 and EN2 in the stratigraphic record was modulated by paleogeographic location,

**Table 1 | Estimates for the age and timescale of Marinoan cap carbonate deposition**

| Radiometric dating | | | | |
|---|---|---|---|---|
| **Age** | **Measurement location** | **Method** | **Geologic formation** | **References** |
| 632.50 ± 0.48 Ma | 5 meters above top of cap carbonate | U-Pb | Doushantuo Formation, China | Condon et al. [6] |
| 632.3 ± 5.9 Ma | 0.9 meters above top of cap carbonate | Re-Os | Sheepbed Formation, Canada | Rooney et al. [56] |
| 635.23 ± 0.57 Ma | Within cap carbonate, 2.3 meters above base | U-Pb | Doushantuo Formation, China | Condon et al. [6] |
| 634.57 ± 0.88 Ma | Base of cap carbonate | U-Pb | Nantuo Diamictite, China | Zhou et al. [4] |
| 636.41 ± 0.45 Ma | 1 meter below base of cap carbonate | U-Pb | Cottons Breccia, Tasmania | Calver et al. [7] |
| 635.21 ± 0.59 M | ~30 meters below base of cap carbonate | U-Pb | Ghuab Formation, Namibia | Prave et al. [8] |
| 635.5 ± 1.2 Ma | ~30 meters below base of cap carbonate | U-Pb | Ghaub Formation, Namibia | Hoffmann et al. [85] |
| **Paleomagnetism** | | | | |
| **Timescale** | **Description** | | **Geologic Formation** | **References** |
| >1.25 Myr | 5 polarity reversals in first 20 meters of cap carbonate | | Mirassol d'Oeste Section, Brazil | Trindade et al. [58] |
| >1.25 Myr | 5 polarity reversals in first 20 meters of cap carbonate | | Terconi Section, Brazil | Font et al. [10] |
| >0.5 Myr | 2 polarity reversals in first 9 meters of cap carbonate | | Jebel Akhdar Section, Oman | Kilner et al. [57] |
| >0.5 Myr | 2 polarity reversals in first 12 meters of cap carbonate | | Second Plain Section, Australia | Schmidt et al. [11] |
| **Sedimentology** | | | | |
| **Timescale** | **Description** | | **Geologic Formation** | **References** |
| $10^3$–$10^4$ yr | Rapid deglaciation and rapid deposition | | Many | Hoffman et al. [1] |
| >$10^5$ yr | Slow deglaciation and slow deposition | | Many | Spence et al. [13] |
| **Astrochronology** | | | | |
| **Age and timescale** | **Measurement location and description** | | **Geologic Formation** | **References** |
| 636.8 ± 0.7 Ma–635.2 ± 0.6 Ma (1.6 Myr) | Base and top of cap carbonate ( ~ 3.9 long eccentricity cycles) | | Doushantuo, Jiulongwan, South China | Sui et al. [24] |
| 635.73 ± 0.45 Ma–634.90 ± 0.43 Ma (0.83 Myr) | Base and top of cap carbonate ( ~ 2 long eccentricity cycles) | | Doushantuo, EYC2 drillcore, South China | This study |
| 636.05 ± 0.43 Ma–634.90 ± 0.43 Ma (1.15 Myr) | Base and top of cap carbonate ( ~ 2.8 long eccentricity cycles) | | Doushantuo, ZK68 drillcore, South China | This study |

(modified from ref. 59). For the depositional timescale of cap carbonates, we adopt the estimates from the Thomas and Catling[59], which are based on paleomagnetic data and sedimentological evidence. The paleomagnetic timescale assumes that one reversal occurred every 250 kyr, consistent with the Miocene, Jurassic, and Cambrian[10]. The sedimentological timescale is derived from broad interpretations of various cap carbonate formations documented in the literature[59].

sedimentation rates, and local environmental factors[67]. This intricate interplay between global events and local factors emphasizes the necessity for precise temporal constraints on the EN1, EN2, and WANCE events. These refined constraints establish a high-resolution chronological framework that is critical for global correlations, elucidating biotic-environmental relationships, investigating carbon cycle dynamics, and defining boundary conditions for geochemical modeling.

## A refined age model for early Ediacaran biotic evolution

Our developed astrochronological framework establishes a refined global age model for early Ediacaran fossil records. The emergence of the Weng'an biota, which hosts diverse and complex multicellular organisms[29], is now astronomically dated to 589.89 ± 0.26 Ma (Fig. 4b). This age is derived from the chemostratigraphic and sequence stratigraphic correlation of its lower boundary with the Sequence 1/2 boundary and the WANCE event nadir in the middle Doushantuo Formation (Supplementary Fig. 31; see Supplementary Note 5 for details)[26]. Our constraint is consistent with, but more precise than, the previously reported age of 587.2 ± 3.6 Ma for its lower boundary[5]. By lithostratigraphic and chemostratigraphic correlation with the ZK68 drillcore, we constrain the age of the Lantian biota, which hosts a diverse assemblage of morphologically differentiated benthic macrofossils[39], between 619.10 ± 0.33 Ma and 582.64 ± 0.42 Ma (Fig. 4b; see Supplementary Notes 5–6 for details). This age range aligns with the proposed nominal age of the lower boundary of the

Lantian biota at ~615 Ma, based on Re-Os dating, with an uncertainty on the order of a few million years[68].

Our framework further refines chronology for key microfossil assemblage zones. Chemostratigraphic and paleontological studies of the Xiangdangping intrashelf basin section indicate that the bases of *Appendisphaera grandis–Weissiella grandistella–Tianzhushania spinose (A.-W.-T.)*, *Tanarium tuberosum–Schizofusa zangwenlongii (T.-S.)*, *Tanarium conoideum–Cavaspina basiconica (Tc-Cb)* and *Tanarium pycnacanthum–Ceratosphaeridium glaberosom (Tp-Cg)* microfossil assemblage zones could be correlated to a horizon 5.0 m above the Doushantuo Member II base, the EP1 zenith, and the EN2 nadir, respectively[26,28] (Supplementary Fig. 31; see Supplementary Note 5 for details). Chemo- and sequence-stratigraphic correlations between the Zhangcunping section and the WD1 and ZK68 drillcores constrain the bases of *A.-W.-T.*, *T.-S.*, *Tc-Cb* and *Tp-Cg* microfossil assemblage zones at 633.64 ± 0.32 Ma, 611.94 ± 0.43 Ma, 589.89 ± 0.26 Ma, and 580.25 ± 0.26 Ma, respectively (Fig. 4b; see Supplementary Note 5 for details).

Our integrated age model for Ediacaran $^{87}Sr/^{86}Sr$ ratios, oceanic oxygenation events, CIEs, and fossil occurrences provides a critical chronological framework for assessing potential causal relationships among these factors. The synchronous occurrence of elevated $^{87}Sr/^{86}Sr$ ratios, CIEs, and oceanic oxygenation events suggests that periodic pulses of oxidant inputs—primarily sulfate and/or evaporites—acted as the unifying mechanism driving extreme negative CIEs[2,66]. These oxidant pulses shaped the redox landscape of the Ediacaran ocean,

**Table 2 | Estimated timing and rates of carbon isotopic change of EN1 and EN2 based on astrochronology results, South China in this study**

| CIE | Onset age (Ma) | δ$^{13}$C$_{carb}$ value at onset (‰) | Age of nadir (Ma) | δ$^{13}$C$_{carb}$ value at the nadir (‰) | Onset change rate[a] (‰/kyr) | End age (Ma) | δ$^{13}$C$_{carb}$ value at the ending (‰) | Recovery change rate[b] (‰/kyr) | Formation |
|---|---|---|---|---|---|---|---|---|---|
| EN1 | 634.64 ± 0.45 | −2.90 | 634.27 ± 0.45 | −13.70 | 0.029189 | 634.06 ± 0.45 | 3.2 | 0.08047 | Doushantuo, EYC2 |
|  | 635.29 ± 0.43 | −3.08 | 634.34 ± 0.43 | −10.71 | 0.008031 | 633.23 ± 0.43 | −3.05 | 0.00690 | Doushantuo, ZK68 |
| EN2 | 581.42 ± 0.26 | 4.90 | 580.25 ± 0.26 | −0.50 | 0.004615 | 580.01 ± 0.26 | 5.23 | 0.023875 | Doushantuo, WD1 |
|  | 584.22 ± 0.43 | −4.67 | 581.57 ± 0.43 | −10.78 | 0.002305 | 581.09 ± 0.43 | −4.06 | 0.014 | Doushantuo, ZK68 |

[a] and [b] represent the rates of carbon isotope change from the start to the nadir of the carbon isotope excursion, and from the nadir to the end of the carbon isotope excursion, respectively.

resulting in pronounced fluctuations between transient oxygenation episodes and more prevalent anoxic periods[69]. Our model reveals significant correlations between these geochemical perturbations and biological innovations. The emergence of new acanthomorphic acritarchs (e.g., *A.-W.-T.*, *Tc-Cb* and *Tp-Cg* microfossil assemblages) and complex macroeukaryotes (e.g., the Weng'an biota) coincides with CIEs and oxygenation events, suggesting a potential causal link between biological innovations and perturbations in biogeochemical cycles[60,70,71]. However, these biological innovations were likely intermittent, punctuated by evolutionary lags or extinctions triggered by sporadic returns to widespread oceanic anoxia, particularly during the early Ediacaran[60,71].

The early Ediacaran evolutionary trajectory is characterized by successive biotic assemblages that developed progressively complex ecosystems over multi-million-year timescales. Despite this ecological advancement, global taxonomic diversity remained relatively stable (Fig. 4b), punctuated by abrupt transitions to novel communities that coincided with episodes of biogeochemical perturbation. This refined perspective on Ediacaran evolution underscores the complex interplay between environmental changes and biological innovations. Geochemical perturbations both spurred evolutionary breakthroughs and posed significant challenges, resulting in a non-linear trajectory of early animal evolution[71,72]. Our high-resolution chronological framework provides a critical tool for exploring and interpreting the key events of the Ediacaran Period, offering new insights into the dawn of animal life on Earth.

## Methods

### Samples and Magnetic susceptibility measurement

Isotopic and geochemical analyses were performed on 214 sedimentary rock samples obtained from three drillcores, including 86 samples from the WD1 drillcore, 60 from the ZK68 drillcore, and 68 from the EYC2 drillcore (Supplementary Data 1). These analyses included measurements of carbonate carbon isotopic composition ($\delta^{13}$C$_{carb}$ and $\delta^{18}$O$_{carb}$), strontium isotope ratios ($^{87}$Sr/$^{86}$Sr), as well as major and trace element concentrations. In the laboratory, samples were cut and selected to exclude visible signs of weathering or veins, and subsequently crushed to particle size of <200 mesh for bulk geochemical analyses.

MS measurements were conducted on the outer, cleaned, round core surfaces at a stratigraphic resolution of ~2 cm using a handheld KT-10 instrument (sensitivity: $1 \times 10^{7}$ SI unit; ZH Instruments, Czech Republic). While the 2 cm sampling resolution may result in relatively few measurements for precession-scale cycles within Member II (Supplementary Figs. 7–8) due to its relatively low sedimentation rates (see the "Estimation of average sedimentation rate" section for details), this does not affect our astronomical calibration which is primarily based on the well-resolved ~405 kyr eccentricity cycles. A total of 1,552 MS data points were collected from the EYC2 drillcore, 5771 from the WD1 drillcore, and 4776 from the ZK68 drillcore (Supplementary Data 2).

### Major and trace elemental concentration analysis

Major and trace element analyses were carried out at the Analytical Laboratory of the Beijing Research Institute of Uranium Geology, China. For major elements, ~1.20 g of sample powder were fused with 6 g lithium tetraborate ($Li_2B_4O_7$) at 1050 °C for 20 min, followed by analysis using an X-ray Fluorescence Spectrometer (AB−104 L). The analytical uncertainty for major element measurements was better than 2.00%. Trace elements were determined using an Agilent 7700e ICP-MS. Sample preparation for trace element analysis involved drying the 200-mesh sample powder at 105 °C for 12 h. Subsequently, 50 mg of the dried powder was digested in a Teflon bomb with 1 ml of $HNO_3$ and 1 ml of HF at 190 °C for over 24 h. After digestion, the sample underwent evaporation and re-dissolution steps, where it was treated with 1 ml of $HNO_3$, 1 ml of Milli-Q (MQ) water, and 1 ml of a 1 ppm

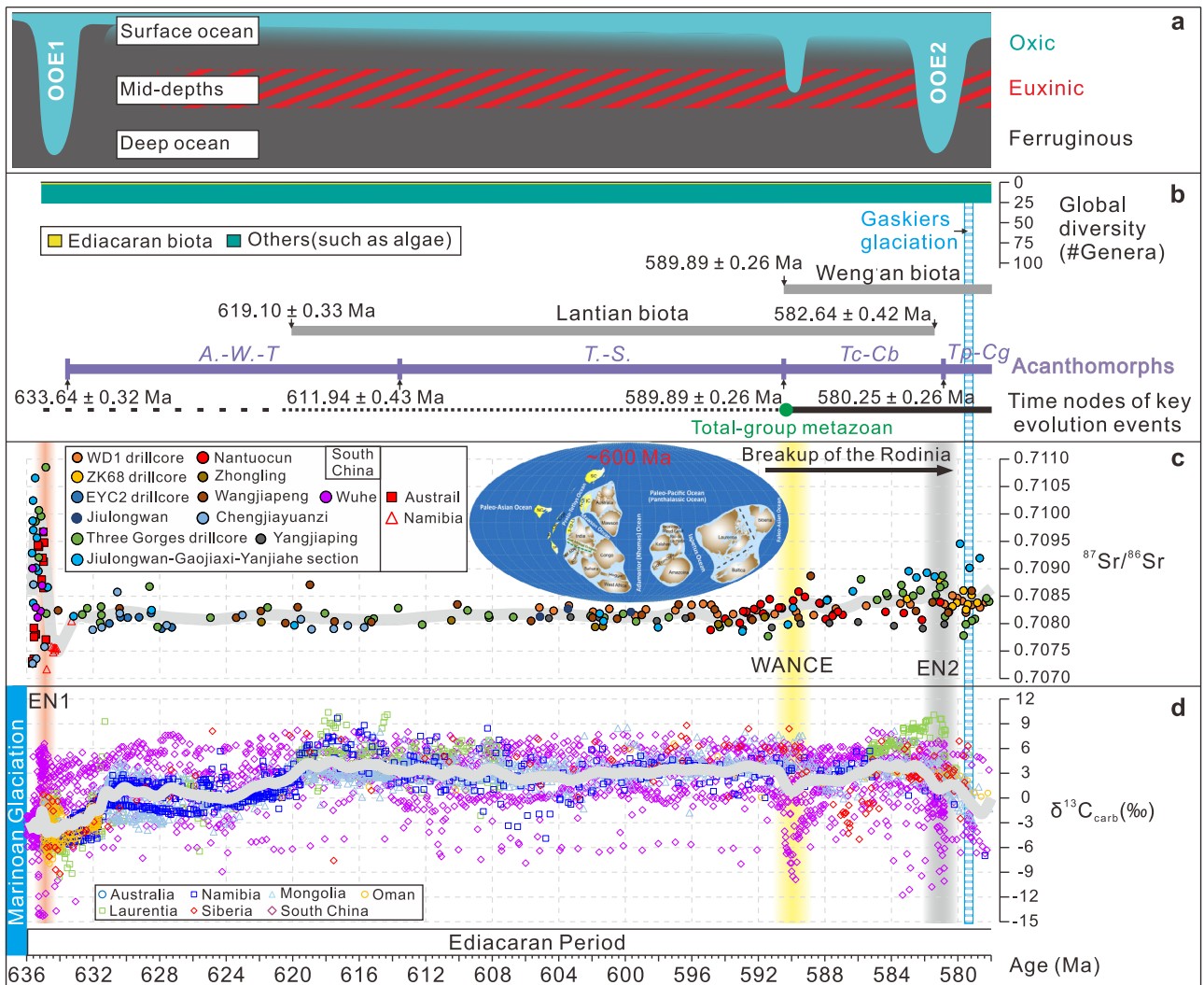

**Fig. 4 | Integrated ocean oxygenation patterns, carbon and strontium isotope curves and their correlation with key fossil records in the early Ediacaran Period. a** Oxygenation pattern for the early Ediacaran ocean based on compiled uranium isotope records and iron speciation data[60,63,71,86]. **b** Global diversity (#Genera), fossil ranges, and the key evolutionary events following ref. 5. See Supplementary Notes 5–6 for a detailed discussion on the age ranges of acanthomorphs, as well as the Lantian and Weng'an biota. **c, d** Compilation of strontium and carbon isotope records for the early Ediacaran Period and their correlation to fossil ranges. Gray line through δ¹³C_carb and ⁸⁷Sr/⁸⁶Sr data denotes locally weighted scatter plot smoothing. Global paleogeographic reconstructions at ~600 Ma illustrate the final stages of the breakup of the Rodinia supercontinent[36]. Age constraints for the $^{87}Sr/^{86}Sr$ ratios and $\delta^{13}C_{carb}$ curves are derived assuming constant sedimentation rates between radioisotopic dates and astrochronological tie points, as detailed in Supplementary Datas 4 and 5. OOE Oceanic oxygenation event; *A.-W.-T. Appendisphaera grandis–Weissiella grandistella–Tianzhushania spinose*; *T.-S. Tanarium tuberosum–Schizofusa zangwenlongii*; *Tc-Cb Tanarium conoideum–Cavaspina basiconica*; *Tp-Cg Tanarium pycnacanthum–Ceratosphaeridium glaberosom*.

indium (In) internal standard solution. The mixture was reheated at 190 °C for over 12 h and then diluted to 100 g with 2% HNO₃ in a polyethylene bottle for ICP-MS analysis.

### Carbonate carbon- and oxygen-isotope analysis

Initially, 60–300 µg of sample powder was dried at 70 °C for 24 h in an argon atmosphere before being loaded into a vial. The samples were then reacted with 100% phosphoric acid under vacuum at 70 °C for 220 s using a Kiel IV device. The resulting CO₂ was subsequently introduced into a MAT 253 isotope ratio mass spectrometer for isotopic measurements. Delta values were calibrated against the international reference standard NBS-19 (δ¹³C = +1.95‰; δ¹⁸O = − 2.20‰) and the Chinese national standard GBW04416 (δ¹³C = +1.61 ± 0.03‰; δ¹⁸O = 1.59 ± 0.11‰). Carbon and oxygen isotope data for carbonates are reported relative to the Vienna Pee Dee Belemnite (VPDB), with a precision better than ± 0.1‰ based on duplicate analyses of GBW04416 and the study samples.

### Strontium-isotope analysis

Guided by detailed petrographic observations and considering the requirements for high Sr content (Sr>200 ppm) and a Mn/Sr ratio <1, a total of 57 samples were selected for strontium isotope analysis to characterize the primary isotopic composition of seawater.

The sample powders (200 mesh) were dried at 105 °C for 12 h. Subsequently, 50-200 mg of powder was accurately weighed into a Teflon bomb, mixed with 1–3 ml each of HNO₃ and HF. The mixture was heated in a stainless steel pressure jacket at 190 °C for over 24 h. After cooling, the sample was evaporated to near dryness at 140 °C, treated with 1 ml of HNO₃, evaporated again, and dissolved in 1.0 ml of 2.5 M HCl. The resulting supernatant was then loaded onto an AG50W resin column for ion exchange. The column was rinsed with 20 mL of 2.5 M HCl to remove matrix elements. The Sr fraction was eluted with 10 mL of 2.5 M HCl and evaporated. The rare earth element (REE) fraction was eluted with 10 mL of 4.0 M HCl after rinsing with 10 mL of 4.0 M HCl. Neodymium (Nd) was subsequently separated from the REE

fraction using the Nd-column method. For further Sr purification, the Sr fraction was converted to a 3 M $HNO_3$ medium and loaded onto Sr-specific resin (SR-B50-S) pre-conditioned with 6 M HCl and 3 M $HNO_3$. After rinsing with 3 M $HNO_3$, Sr was eluted using MQ $H_2O$ and evaporated to dryness for mass spectrometric measurement.

Strontium isotope analyses were performed using a Neptune Plus MC-ICP-MS (Thermo Fisher Scientific, Dreieich, Germany) at the Analytical Laboratory of Beijing Research Institute of Uranium Geology, Beijing, China. The analytical standard NIST SRM 987 was used, with a sample-standard bracketing technique employed to enhance measurement reproducibility. All $^{87}Sr/^{86}Sr$ ratios were normalized to the NIST SRM 987 value of 0.71025. For more details, please refer to ref. [65].

## $^{87}Sr/^{86}Sr$ ratio and magnetic susceptibility as paleoclimate proxies

The long-term trend in seawater strontium isotopes is controlled by the balance between two primary sources: radiogenic $^{87}Sr$ from continental weathering and less radiogenic $^{87}Sr$ from mantle-derived materials at mid-ocean ridges[73]. Intensified continental weathering increases the riverine input of radiogenic $^{87}Sr$ into the ocean, thereby potentially elevating seawater $^{87}Sr/^{86}Sr$ ratio[74]. This relationship makes the seawater $^{87}Sr/^{86}Sr$ ratio a valuable proxy for assessing long-term continental weathering intensity and associated climate changes[73].

MS data measures the concentration of magnetic minerals in sediments[75]. It is widely used as a proxy for detrital fluxes from terrestrial sources, primarily transported via fluvial processes in marine environments[75,76]. Aeolian inputs can also contribute to MS signals, particularly during glacial periods. However, during the Gaskiers interval, multiple continental paleoclimate proxies (e.g., MS, $^{87}Sr/^{86}Sr$ ratios, carbonate-clumped isotope paleotemperature reconstructions and the Chemical Index of Alteration; Supplementary Fig. 30) show notable decreases. Combined with the sedimentological context, these observations strongly support a predominantly fluvial origin for the MS variations in this study. Numerous studies have demonstrated that MS is one of the most reliable indicators of astronomical cycles in sedimentary records making it a powerful tool (e.g., ref. [77]).

## Time series methods

Dominant wavelengths and potential astronomical frequencies were examined using multi-taper method (MTM) power spectra[78], with 90%, 95%, 99%, and 99.9% confidence levels against robust AR(1) red noise models[79]. To address the issue of multiple testing in cyclostratigraphy, we applied a more stringent approach by testing power spectral peaks against the 5% false discovery rate (FDR) threshold[80]. This method provides a more restrictive criterion for identifying significant spectral components. Evolutive Fast Fourier Transform (eFFT) with a sliding window was applied to inspect dominant frequency changes due to sedimentation rate variations[75]. Average sedimentation rates were derived through integrated analysis of independent age constraints and TimeOpt statistical modeling[43,44]. These sedimentation rates, combined with theoretical Milankovitch frequency ratios characteristic of the early Ediacaran Period[41,42] (detailed in Supplementary Note 2), enabled the attribution of identified dominant frequencies to specific astronomical cycles. The astronomical signals were extracted using Gaussian bandpass filtering[75]. To further evaluate the reliability of our cyclostratigraphic interpretation, the amplitude modulation of a bandpass signal was analyzed by applying the Hilbert transform to extract the signal's envelope in the depth domain[81]. MTM spectral analysis of the envelope was then conducted to identify the amplitude modulation cycles of the shorter-wavelength signal. Given that eccentricity modulates precession to drive climate change, the Hilbert transform offers a robust method to test for such diagnostic spectral patterns[48].

Astronomical tuning was performed using the long eccentricity cycle as a primary target, implemented through the 'Age Scale'

function in Acycle, facilitating the transformation from depth to time domain and generating a floating ATS. The observed cycle periods were subsequently compared to the predicted estimates—~405 kyr (long eccentricity), 95.2–132.2 kyr (short eccentricity), ~30.6 kyr (obliquity), and ~16.5 kyr (precession)—for the early Ediacaran (~610 Ma), as derived from solar system evolution models[41,42]. This floating ATS was subsequently calibrated against high-precision radioisotopic dates to establish an absolute ATS.

TimeOpt is a statistical method for optimizing sedimentation rate estimation through orbital signal analysis[43,44]. This approach converts proxy series from depth to time domain at various test sedimentation rates and employs Taner filtering, and Hilbert transforms to isolate potential precession cycles. The method generates two correlation coefficients: $r^2_{envelope}$ (correlation between the amplitude envelope and theoretical eccentricity) and $r^2_{power}$ (correlation with combined eccentricity-precession frequencies). The product of these coefficients ($r^2_{opt} = r^2_{envelope} \times r^2_{power}$) determines the optimal sedimentation rate, with the highest $r^2_{opt}$ value indicating the most probable rate. Statistical significance is assessed through Monte Carlo simulations using red noise series, with $p$-value < 0.05 considered statistically significant. The MTM, eFFT analyses, Gaussian bandpass filters, Hilbert transform and Age Scale were performed using Acycle v2.8[82]. The TimeOpt analyses were conducted using with "Astrochron" (https://search.r-project.org/CRAN/refmans/astrochron/html/astrochron-package.html; ref. [83]) in R[84].

TimeOpt analysis was used to refine sedimentation rate estimates for each subset, using initial constraints derived from eFFT analysis, radiometric dating, and geological context. While eFFT analysis provides a broad framework for potential sedimentation rates, TimeOpt identifies the most likely rates by maximizing the correlation between observed stratigraphic cycles and theoretical Milankovitch periodicities ($r^2_{opt}$). The sedimentation rate ranges tested in TimeOpt were chosen to balance geological realism with computational efficiency, focusing on rates consistent with radiometric age constraints and the depositional context of the Doushantuo Formation (e.g., intrashelf basin sedimentation rates of ~0.2–0.5 cm/kyr; ref. [24]). These geological considerations ensured that the tested ranges aligned with established knowledge of comparable depositional environments. In the EYC2 drillcore, the TimeOpt analysis of the untuned MS series indicates that the most likely mean sediment accumulation rate is 0.35 cm/kyr for the entire studied stratigraphic interval, at which the null hypothesis significance level of no orbital forcing is 0.062 (Supplementary Fig. 10). In the WD1 drillcore, the TimeOpt results indicate the following optimal mean sedimentation rates: 0.37 cm/kyr for subset D2-1, 0.35 cm/kyr for subset D2-2, 0.49 cm/kyr for subset D2-3, 0.37 cm/kyr for subset D2-4, and 0.65 cm/kyr for subset D3, at which the confidence level of orbital forcing is 99.4%, 98.4%, 94.75%, 97.9%, and 98.1%, respectively (Supplementary Figs. 11–15). The TimeOpt analysis of untuned MS series in the ZK68 drillcore reveals the most likely mean sediment accumulation rates as follows: 0.66 cm/kyr for subset D1, 0.16 cm/kyr for subset D2-1, 0.18 cm/kyr for subset D2-2, 0.10 cm/kyr for subset D2-3, 0.14 cm/kyr for subset D2-4, 0.15 cm/kyr for subset D2-5, and 0.53 cm/kyr for subset D3, at which the null hypothesis significance level of no orbital forcing is 0.005, 0.0235, 0.078, 0.064, 0.005, 0.041 and 0.04, respectively (Supplementary Figs. 16–22). Besides, we conducted supplementary TimeOpt analysis using expanded sedimentation rate ranges for two representative intervals (subsets D2-4 and D3 from the WD1 drillcore), allowing for sedimentation rates significantly higher and lower than those initially tested. The results show that the optimal sedimentation rates remained consistent with our original estimates, underscoring the robustness of our approach (Supplementary Figs. 14, 15, 23 and 24). Sedimentation rates outside the originally selected range—either significantly higher or lower—produced lower $r^2_{opt}$ values and/or $P$-values that did not meet statistical significance (Supplementary Figs. 14, 15,

23 and 24). These outcomes support the validity of our original sedimentation rate selection, as rates within the initially selected range consistently produced the highest $r^2_{opt}$ values and statistically significant *P*-values, capturing the most plausible sedimentation rates.

### Estimation of average sedimentation rate

A CA-ID-TIMS U-Pb age of 634.57 ± 0.88 Ma was reported for the ES-1 ash bed from the topmost Nantuo Formation at Eshan in eastern Yunnan Province, South China[4]. A U-Pb concordia age of 635.23 ± 0.57 Ma was obtained from an ash bed located 2.3 m above the base of Member II at the Wuhe-Gaojiaxi section, South China[6], suggesting that the Member I-II boundary is slightly younger than ~635.23 Ma. The Gaskiers glaciation is constrained between 579.63 ± 0.15 Ma and 579.24 ± 0.17 Ma based on CA-ID-TIMS dating from Newfoundland[35]. A high-resolution paleotemperature record from the ZK312-P312 drillcore in the Yangtze Gorges area indicates a maximum secular cooling ~15 m above the boundary between Member II and Member III, which is interpreted as the peak expression of the Gaskiers glaciation. Consequently, the Member II/III boundary of the Doushantuo Formation is estimated to be slightly older than ~580 Ma[31].

These datings provide estimated durations of ~0.79 Myr for Member I[4], ~55.2 Myr for Member II and ~0.39 Myr for the Gaskiers glaciation within Member III. Based on these durations, we calculated average sedimentation rates for different intervals. Member I has thicknesses of 2.5 m and 7.51 m in the EYC2 and ZK68 drillcores, respectively, yielding estimated average sedimentation rates of ~0.32 cm/kyr and 0.95 cm/kyr. The Member II interval in the ZK68 drillcore, spanning from 1471.99 m to 1399.22 m (with a thickness of 72.77 m), has an estimated average sedimentation rate of ~0.13 cm/kyr. For the Gaskiers glaciation interval, sedimentation rates vary between the two studied drillcores. In the WD1 drillcore, the interval from 2055.44 m – 2057.68 m (with a thickness of 2.24 m) yields an approximate sedimentation rate of 0.57 cm/kyr. In contrast, the ZK68 drillcore shows a slightly lower rate of ~0.41 cm/kyr for the interval from 1389.1 m to 1390.72 m (with a thickness of 1.62 m). These observations suggest that the sedimentation rate in the lower part of Member III in both studied drillcores may be slightly higher than the average rates calculated for the Gaskiers glaciation interval.

Further chronological control is provided by the WANCE event, its nadir dated at 587.2 ± 3.6 Ma based on Re-Os dating[5] (Fig. 2). This horizon correlates with the nadir of a WANCE ~ 32.2 m below the Member II/III boundary in the WD1 Drillcore within the intrashelf basin setting. This correlation implies a duration of ~7.2 Myr for the interval spanning from the nadir of the WANCE event at 2099.5 m to the top of Member II at 2067.26 m in the WD1 drillcore, with a thickness of 32.24 m and an estimated average sedimentation rate of ~0.45 cm/kyr for the upper part of Member II of the Doushantuo Formation.

At the Wuhe-Gaojiaxi section, an ash layer 2.3 m above the base of the Doushantuo Formation (with Member I totaling 4 m) yields an age of 635.2 ± 0.6 Ma, while at the Jijiawan section, an ash layer 5 m above the top of Doushantuo Member I is dated to 632.5 ± 0.5 Ma[6]. These U-Pb dates provide crucial chronological constraints, allowing for an estimated average sediment accumulation rate of ~0.25 cm/kyr for the lower Doushantuo Formation. The WD1 and EYC2 drillcores are paleogeographically proximal to the Wuhe-Gaojiaxi and Jijiawan sections, suggesting that their sedimentation rates likely fall within a similar range. Considering the previously calculated rates for upper Member II of the Doushantuo Formation, including the higher estimates, it is reasonable to infer that the average sedimentation rate for Doushantuo Member II in the WD1 and EYC2 drillcores likely falls within a broader range of ~0.25–0.45 cm/kyr. The sedimentation rates derived from independent age data show broad consistency with TimeOpt results for the WD1, ZK68, and EYC2 drillcores.

## Data availability

All data used in this study are provided in the Supplementary Information. Source data are provided with this paper.

## Code availability

The Acycle[82] software used in this study is available for download at https://github.com/mingsongli/acycle. Astrochron[83] package can be accessed from https://cran.r-project.org/web/packages/astrochron/index.html. Additional code and configuration files relevant to the analyses are provided in the Supplementary Information.

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

## Acknowledgements

We thank Stephen R. Meyers for his insightful review of an early draft of the manuscript and his assistance with the TimeOpt analysis. This project was funded by the National Natural Science Foundation of China (grant nos. 42302129 (T.Z.), 42488201 (C.M.), U24B6001 (T.F.), 42130208 (C.L.), 42425002 (C.L.), 42172137 (C.M.), and 42272173 (Y.L.)) and Sichuan Science and Technology Program. (2023NSFSC1986 (C.M.)), and the Key Laboratory of Sedimentary Basin and Hydrocarbon Resources of the Ministry of Natural Resources Open Fund (cdcgs2023004 and cdcgs2023006 (C.M.)). This study is supported by the High-performance Computing Platform of Chengdu University of Technology and a contribution to International Geoscience Programme (IGCP) Project 739 and the Deep-time Digital Earth (DDE) Big Science Program.

## Author contributions

C.M. designed the study. T.Z. and C.M. performed the study. T.Z., Q.G., M.K., and W.L. collected samples and measured MS data. T.Z., C.M., C.L., A.-C.D.S., M.L., Y.L., T.F., and M.H. analyzed and interpreted data. T.Z. and C.M. wrote the paper with input from all authors.

## Competing interests

The authors declare no competing interests.
