## [Transparent Peer Review file · Nature Communications]

Astronomically calibrating early Ediacaran evolution

Corresponding Author: Professor Chao Ma

Version 0:

Reviewer comments:

Reviewer #1

(Remarks to the Author)

Dear authors, dear editor,

The reviewed work presents an impressive effort of an almost 60 Myr spanning anchored astrochronology of the early Ediacaran. It is, to my knowledge, the first astrochronology of this length in this time period. As such, the improved time constraints that this manuscript provides will be of use to any early Ediacaran worker. I am not an expert on the Ediacaran, having worked mostly on Paleozoic cyclostratigraphy, so my knowledge of Ediacaran stratigraphy is limited. However, having consulted some literature on Ediacaran events and previous stratigraphic timescales for this interval, I believe that this work addresses several issues that were previously present in constructing an early Ediacaran timescale. I also appreciate the extensive documentation and data supplied by the authors.

I have answered the Nature Communications questions below. Overall, I have only minor suggestions, which are given on a line-by-line basis in the last section. Finally, I have a few very minor comments given as annotations to the PDFs themselves. As I believe that the authors can address these with ease, my recommendation is to publish this work after minor revisions.

Kind regards,

Nina Wichem

General questions:

What are the noteworthy results?

The presented work is a very impressive, almost 60 Myr long anchored astronomical timescale for the early Ediacaran, a period for which as of yet no such detailed chronostratigraphy has been established. It covers key events in the Ediacaran, including two carbon isotope excursions that are often used as correlative chemostratigraphic markers (EN1 and EN2), the Marinoan deglaciation, as well as important evolutionary shifts such as the Lantian biota.

Will the work be of significance to the field and related fields? How does it compare to the established literature? If the work is not original, please provide relevant references.

I consider this work significant not only within the cyclostratigraphic community, of which it is one of the first and certainly the most extensive work in this period of geologic time, but also for any researcher working in the early Ediacaran that relies on age constraints and durations. This work is especially valuable in this aspect for two reasons. 1) It places two key Ediacaran carbon isotope excursions in an astrochronological framework. These excursions have been recognized in various sections across the globe, allowing this astrochronology to be correlated to other sections. 2) From what I have read as a non-Ediacaran researcher, biostratigraphy is of limited use in this interval due to strong taphonomic biases, which places more importance on the remaining stratigraphic approaches.

Does the work support the conclusions and claims, or is additional evidence needed?

Yes; as far as I can tell, the interpretations of the presented data are robust. The fact that similar results are obtained from both shallow and deeper water sections supports the cyclostratigraphic interpretations.

Are there any flaws in the data analysis, interpretation and conclusions? - Do these prohibit publication or require revision?

No, I have not come across any flaws in analysis, interpretation, or conclusions.

Is the methodology sound? Does the work meet the expected standards in your field?

I will mostly comment on the cyclostratigraphy part, as that is my area of expertise. The methodology is in line with state-of-the-art cyclostratigraphic approaches. The authors go beyond the statistical methods of conventional cyclostratigraphic studies (which have rightfully been criticized in recent years) and apply a more statistically rigorous method, the False Discovery Rate (FDR) approach. The supplementary text on potential diagenetic alteration is also thorough.

Is there enough detail provided in the methods for the work to be reproduced?

Yes. I commend the authors for their thoroughness in reporting the methodology, as well as the supporting data and supporting code. The amount of documentation associated with this study should be the standard for cyclostratigraphic studies.

Detailed comments:

Comments on the main article:

Abstract in general: While the abstract summarizes the main findings well, I miss two things: 1) a more explicit mention of the location of the studied records; and 2) maybe a mention of the Gaskiers glaciation. While the well-dated Gaskiers glaciation is not recorded in South China apart from a $\delta^{18}\text{O}$ -inferred cooling phase, it is indirectly integrated in the astrochronology here. From reading about Ediacaran stratigraphy, I got the sense that the Gaskiers glaciation is a key stratigraphic marker, especially if it can be reliably linked to EN2, so it might be worth mentioning this explicitly in the abstract and introduction. I say 'maybe' because I am not an Ediacaran worker, so ultimately the authors might be a better judge of its relevance.

Presentation of the results in general: I find it useful that the authors have gathered the main dates and duration constraints they arrived at in a table (Table 2). The figures are clear. The figure text is often a bit small, but I assume that this comes from the graphic limitations of the review file format.

Line 123: Regarding the continuous deposition of the Doushantuo Fm, I came across this recent paper: Li et al. 2024, EPSL, <https://doi.org/10.1016/j.epsl.2024.118715>. It poses that there is a major unconformity somewhere in Member III of the Doushantuo Fm. This is stratigraphically above the studied cores, so it should not affect the presented astrochronology. However, since the manuscript also discusses Member III and it is shown in the correlation panels, it might be worth it to explicitly address this.

Line 174: "These [11.5 Myr, 5.5 Myr, 2.4 Myr, 405 kyr, 141-90 kyr, 32.9-28.3 kyr and 21.3-15.3 kyr] peaks align with major periods predicted for orbital forcing of early Ediacaran solar insolation [Waltham 2015]". To which major periods do the 11.5 and 5.5 Myr cycles correspond? I could not find that in the Waltham citation or the Waltham calculator.

Line 370: The MS resolution of 2 cm implies that some precession cycles have very few (<5) measurements per cycle. I don't think this hinders the astrochronological interpretation, which is based on the 405-kyr cycle, but it should perhaps be stated somewhere for transparency.

Line 425: It might be good to add that the authors implicitly assume that MS mostly reflects a fluvial, rather than an aeolian signal. I infer this from the usage of decreases in MS (and $87\text{Sr}/86\text{Sr}$) as an indicator for the onset of the Gaskiers glaciation (Supplementary materials lines 381-383). An aeolian MS signal would be expected to increase during a glacial period – although this drying signal may not be as significant in a world without land vegetation.

Comments on the supplementary materials:

Line 51: The name 'Shuram excursion' is not mentioned in this paragraph as a probable synonym for EN3, but it is mentioned in Supplementary Figure 2. I would add it in this paragraph for completeness; from my (limited) knowledge of Ediacaran isotope stratigraphy, it is a commonly used term.

Line 187-193: This phrasing makes it sound as if the presence of astronomical cycles is assumed as given. So, once you

know the sedimentation rates, you can assign any frequency that fits to the right astronomical parameter. But we do not know whether a certain sedimentary succession has preserved astronomical cycles to such a degree that they can be extracted. I think there is plenty additional evidence that these are indeed astronomical cycles, as is shown later in the manuscript: e.g., the 405-kyr signal shows a clear 2.4 Myr amplitude modulation; MS is a proxy that can and has been shown to record all these astronomical parameters in similar environments; and also the sheer length of the record implies that it would be stranger to NOT detect any astronomical signal whatsoever within this interval. It should just be stated a bit more explicitly that the authors are not blindly assuming the presence of astronomical signals (which I think they certainly do not).
Supplementary Note 3 in general: Regarding sedimentation rate changes (which I think are addressed well, cyclostratigraphy-wise), I miss a discussion on how these relate to lithological changes and whether the sedimentation rate changes make sense in that context.

Line 2736 (Supplementary Fig. 23): For visual clarity, it would be nice to mark the identified astronomical parameters on the eFFT plots here, similar to how it was shown in Supplementary Fig. 9c.

Reviewer #2

(Remarks to the Author)

Dear Isabelle,

Please find below my review of the ms "Astronomically calibrating early Ediacaran evolution" by Zhang and others for Nature Communications. They produce in one go an astrochronology for almost 60 million years of early Ediacaran history, which would be a truly amazing achievement by itself. The short ms reads overall fine, but the cyclostratigraphic approach, results and inter-pretations are only briefly summarized in the ms and are being dealt with in detail in the SI. For this reason I mainly focused on that part of the SI. There are in my opinion some major issues with the approach that the authors have to clarify and that currently prevent acceptance of the ms. These are given below. However, this criticism is not meant to downgrade the importance of the ms, I just want to be more certain that the succession is continuous and why the authors decided to follow a specific statistical approach while in my opinion an alternative approach might have been more logical:

1) Cyclostratigraphic analysis in SI. As stated before the important details of the cyclostratigraphic analysis are "hidden" in the SI which has the risk that readers will not pay that much attention to it, while these details are of critical importance for the ms. But this is not for the authors to blame, but in my opinion a general problem with many papers today. Now only a short summary is given in the ms itself and a single review figure with the important results of bandpass filtering in the depth and time domain. Unfortunately this important figure was rather difficult to view as it is somewhat blurred in the pdf.

2) Very low sedimentation rates: This is not a point of critics on the cyclostratigraphy but the authors have to clarify in the ms why the sedimentation rates (of less than 1 cm/kyr) are so low and whether the succession is essentially continuous on the Milankovitch scale. Is that normal? My experience is that sedimentation rates are generally higher. The risk exists that the succession contains hidden unconformities and hiatuses so that the average sedimentation rate becomes lower. This principle is outlined among others in Andrew Miall papers (his Sedimentation Rate Scale (SRS)). The authors have to provide arguments for the continuity of the studied succession. However, most if not all of the Doushantuo Fm is deeper marine so will be much less prone to erosion and might therefore well be continuous and also explain the low sedimentation rate. But since it would be such an amazing succession, this has to be clarified in the main text.

3) The authors split up the succession in distinct intervals with specific sedimentation rates. However, this approach assumes abrupt changes in sedimentation rate which is less likely. The selection of the intervals are based on ESA but this is a windowed approach (they used quite long windows) that will not provide clear evidence for such abrupt changes.

4) TimeOpt. The authors subsequently use TimeOpt with constant sedimentation rates to compute the optimal sedimentation rate for each of the previously selected intervals. However, they use a very short range of possible sedimentation rates that only covers the expected rate based on the eFFT analysis (?). In my opinion, they should use a much wider range to allow other possible sedimentation rates. This is especially important in view of point 1. However, their argument might be that they already used that wider range already in the eFFT analysis, but eFFT does not provide estimates of the most likely (optimal) sedimentation rate. They base their interpretation on the their own interpretation of the cycles ratios under changing sedimentation rates.

5) eTimeOpt and eCOCO. So my question is why the authors chose for the combination of eFFT and TimeOpt and did not select continuous evolutive techniques such as eTimeOpt and/or eCOCO. Especially eCOCO, which like eTimeOpt and eFFT start from variable sedimentation rates and not a constant rate like TimeOpt. Of course they used eFFT as a first step to subdivide the record, but eCOCO will provide optimal sedimentation rates per window, so in fact it combines the eFFT with TimeOpt. There is also the option in one of the programs to trace for instance the expression of the long eccentricity cycle to create an astrochronology and this avoids the necessity to use selected intervals with constant sedimentation rates. This may provide an alternative for the bandpass filtering of the cycle that is supposedly related to long eccentricity and shown in the summary figure in the ms, as it allows for variable sedimentation rates that constant sedimentation rates per fixed interval. The authors again have to show why they selected their approach of combining eFFT with TimeOpt, and not use eTimeOpt and/or eCOCO.

Reviewer #3

(Remarks to the Author)

This manuscript reports on a new high-resolution, astronomically-calibrated chronostratigraphy for the early Ediacaran based on MS cyclostratigraphy from Chinese drill cores. The work is highly significant as there is very little in the literature about the chronostratigraphy for this time period. It provides detailed time calibration for significant events in Earth history in this critical time period, i.e. the timing of the Gaskiers glaciation and the duration of the cap carbonate for the end of the Marinoan global glaciation (snowball Earth event), as well the timing and global correlation of the evolution of early multi-cellular life. This is an extremely significant result for Earth history and the early evolution of life on the planet.

The cyclostratigraphy that is the major contribution of this work is a highly robust result. The authors first derived average sediment accumulation rates from radiogenic dating and were able to identify Milankovitch peaks in the MS cyclostratigraphy. These peaks are all significant at the 95% or greater significance level. The astronomically-calibrated chronostratigraphy that results from this careful work is high quality.

The paper is well-written and logically organized. The supplemental information is quite detailed and extensive, but important to have available to reader.

The only scientific comment I have is not a criticism and should not hold up publication of this important work. The sediment accumulation rates reported are all less than 1 cm/kyr which from my experience is somewhat low. I have seen low rates off the US east coast in the Miocene, so rates this low are not unheard of, but given that these sediments were deposited in the Ediacaran when there was no terrestrial vegetation to slow down erosion of the continents it is surprising. Maybe some comment about this finding would be useful for the reader, but certainly not necessary.

K.P. Kodama Oct 28, 2024

Version 1:

Reviewer comments:

Reviewer #2

(Remarks to the Author)

The authors did a very good job in replying to the critical points raised by the reviewers and addressed most of the concerns raised in a quite convincing manner, although this does not - logically so - remove all the critical issues. Yet, in my opinion, the ms can be accepted for publication in Nature Communications after they take some remaining minor comments into account.

Comments:

Low sed rates: Still the low sedimentation rates are exceptional and I advise the authors to be very careful in their wording. This also holds for their argument that radiometric dates provide evidence for this low sedimentation rate because what do these tell us about potential hiatuses or condensed intervals that are present in the succession? In addition the detection of hiatuses is not always straightforward, also or especially not in cores.

Wider range of sed rates: The authors did a good job in providing examples of running TimeOpt over a wider range of sedimentation rates. It might be a good idea to include these in the SI, although I leave that to the discretion of the authors. It is clear that the authors experimented with different statistical techniques before developing and choosing their preferred approach.

(e)COCO: I agree with the authors that the use of eCOCO may not be recommended as long as the theoretical shortcomings are not repaired. It is very good to see that they contacted Steve Meyers on such critical issues. However, despite these shortcomings, the somewhat confusing eCOCO results likely also tells something about the overall quality of their data and how straightforward - or not - these data can be interpreted in terms of Milankovitch forcing. Moreover, one of the problems with cyclostratigraphy is that often more cycle period ratios will closely fit the ratios of the astronomical cycles, resulting in more than one possible outcome / interpretation. For this reason, an integrated stratigraphic approach in particular using tools like radio-isotopic dating is critical and it is this approach that the authors largely follow (but see above).

**Dear Reviewers,**

We are particularly grateful for the constructive comments and insightful
suggestions provided by the reviewers, which have significantly enhanced the quality
and clarity of our manuscript.

In response to the reviewers' feedback, we have thoroughly addressed all the
comments and carefully revised the manuscript accordingly. To facilitate the review
process, all modifications are highlighted **in red text** for easy reference. Below, we
provide a point-by-point response to each reviewer's comments and detail the major
changes we have made to improve our work. References to line numbers in our
responses correspond to the **tracked-changes version** of our manuscript and
supplementary materials.

We hope that our revisions meet the expectations of the reviewers and improve the
overall impact of our work. Thank you again for your thoughtful and constructive
feedback, and we look forward to your further evaluation of our manuscript.

Best regards,

Chao Ma

Professor

State Key Laboratory of Oil and Gas Reservoir Geology and Exploitation & Institute
of Sedimentary Geology, Chengdu University of Technology

No.1 East 3rd Road, Erxianqiao, Chenghua District, Chengdu 610059, China

Email: machao@cdut.edu.cn

**Response to Dr. Nina Wichern (Reviewer 1)**

**We are sincerely grateful to Dr. Nina Wichern for the thorough review and**
**positive assessment of our manuscript. Your precise summary of our work**
**demonstrates a clear understanding of our research objectives and findings. We**
**appreciate your constructive feedback and have addressed each of your comments**
**and suggestions in detail below.**

**Response to Comments on the main article:**

***Comment 1:** Abstract in general: While the abstract summarizes the main findings well,*
*I miss two things: 1) a more explicit mention of the location of the studied records; and*
*2) maybe a mention of the Gaskiers glaciation. While the well-dated Gaskiers*
*glaciation is not recorded in South China apart from a $\delta^{18}O$ -inferred cooling phase, it*
*is indirectly integrated in the astrochronology here. From reading about Ediacaran*
*stratigraphy, I got the sense that the Gaskiers glaciation is a key stratigraphic marker,*
*especially if it can be reliably linked to EN2, so it might be worth mentioning this*
*explicitly in the abstract and introduction. I say ‘maybe’ because I am not an Ediacaran*
*worker, so ultimately the authors might be a better judge of its relevance.*

**Reply:** We appreciate this valuable suggestion to enhance the clarity and context
of our abstract. We fully agree that providing more explicit geographical
information and discussing the Gaskiers glaciation would strengthen our paper's
accessibility and broader implications. In response to these constructive comments,
we have made the following revisions:

**First, we have added "anchored by the radioisotopic date of the Gaskiers**
**glaciation onset, based on key sections from South China."** to provide clear
**geographical context for our studied records. See Lines 33-35 in the main text.**

**Second, we have expanded the discussion of the Gaskiers glaciation and its**
**relationship to EN2 in the introduction by adding “Despite these challenges, the**
**complete shallow-to-deep water sedimentary sequences in South China, coupled with**

*abundant carbon isotopic data, fossil records, multiple radiometric dates and well-*
*established stratigraphic frameworks, make it an exceptional region for studying*
*both Ediacaran cyclostratigraphy and the co-evolution of life and environments (refs.*
*2–6,26–29). Notably, recent studies have identified the EN2 CIE in both South China*
*and Newfoundland, precisely constrained to before 580.9 ± 0.4 Ma and immediately*
*preceding the Gaskiers glaciation^{30–32}, provides a critical chronostratigraphic marker.*
*The Gaskiers glaciation, a major climatic event documented across at least eight*
*ancient continents and well-constrained by high-precision radiometric ages^{33–35},*
*presents new potential for establishing a robust Ediacaran astronomical time*
*framework.” See **Lines 93-102** in the main text.*

**Comment 2:** *Presentation of the results in general: I find it useful that the authors have*
*gathered the main dates and duration constraints they arrived at in a table (Table 2).*
*The figures are clear. The figure text is often a bit small, but I assume that this comes*
*from the graphic limitations of the review file format.*

**Reply:** *Thank you for these positive comments on our data presentation. We are*
*pleased that Table 2 effectively summarizes our chronological results and that the*
*figures were found to be clear. Regarding the figure text size, we have carefully*
*checked all figures and adjusted the text size in the Figures 1-4 to ensure optimal*
*readability. See **Lines 573, 583, 614 and 631** for the revised Figures 1-4 in the main*
*text.*

**Comment 3:** *Line 123: Regarding the continuous deposition of the Doushantuo Fm, I*
*came across this recent paper: Li et al. 2024, EPSL,*
*<https://doi.org/10.1016/j.epsl.2024.118715>. It poses that there is a major unconformity*
*somewhere in Member III of the Doushantuo Fm. This is stratigraphically above the*
*studied cores, so it should not affect the presented astrochronology. However, since the*
*manuscript also discusses Member III and it is shown in the correlation panels, it might*
*be worth it to explicitly address this.*

**Reply:** Thank you for bringing up this important point and for referring us to the
recent paper by Li et al. (2024). This study raises an intriguing hypothesis
regarding a potential major unconformity within Member III of the Doushantuo
Formation, with a proposed >44 Myr gap at the boundary between Members II
and III. While this proposed hiatus is stratigraphically above the interval studied
in our manuscript, it is indeed relevant to address given that Member III is
discussed and shown in our correlation panels.

Our observations and analyses suggest an alternative interpretation. Detailed
examination of WD1 and ZK68 drillcores, we observe continuous sedimentary
successions through the studied interval, with no evidence of significant
depositional breaks near the Members II/III boundary (See **Response-Figure 1**).
If a regional or global hiatus spanning >44 Myr existed at the Members II/III
boundary, one would expect to observe prominent unconformity surfaces,
particularly in shallow-water areas (e.g., inner shelf on the Yangtze Platform),
rather than the subtle facies changes described in their paper. These lithofacies
variations are commonly interpreted as the result of sea-level fluctuations.

Furthermore, existing sedimentological and sequence stratigraphic studies
have not documented widespread major unconformities at the Members II/III
boundary in either the Yangtze platform intrashelf basin or the inner shelf
environments, where the Doushantuo Formation consistently maintains a
thickness of ca. 100-180 m, which is much greater than the ~33 m thickness in their
studied section (e.g., Jiang et al., 2011; Zhu et al., 2013). In contrast, many drillcores
from the deep-water slope/basin setting in eastern Guizhou, show reduced
formation thickness (ca. 30-60 m) with clear regional concentration (**Response-**
**Figure 2**). This pattern more likely reflects the influence of paleotopographic highs
experiencing localized depositional limitations during sea-level lowstands (e.g., the
Gaskiers glaciation), rather than a basin-wide hiatus of >44 Myr.

To address this point explicitly, we have revised the manuscript to include the
following statement “*Although a recent study proposed a potential stratigraphic*
*hiatus near the boundary between Members II and III³⁸, our detailed core*

*examination reveals no sedimentological evidence for such a discontinuity (Fig. 1c,*
 *d).” See Lines 134-136 in the main text.*

 **Response-Figure 1. Core photographs showing facies variation across the**
 **Members II-III boundary, Doushantuo Formation from ZK68 and WD1 drillcores.**

Zk101 drillcore-Eastern Guizhou Province

Zk1320 drillcore-Eastern Guizhou Province

Zk3312 drillcore-Eastern Guizhou Province

**Response-Figure 2. Thickness Variations of the Doushantuo Formation in Eastern**
**Guizhou of South China based on drilling data.**

**Comment 4: Line 174: “These [11.5 Myr, 5.5 Myr, 2.4 Myr, 405 kyr, 141-90 kyr, 32.9-**
**28.3 kyr and 21.3-15.3 kyr] peaks align with major periods predicted for orbital forcing**
**of early Ediacaran solar insolation [Waltham 2015]”. To which major periods do the**
**11.5 and 5.5 Myr cycles correspond? I could not find that in the Waltham citation or**
**the Waltham calculator.**

**Reply:** Thank you for this careful observation regarding the long-period cycles.
You are correct that these specific periods (11.5 Myr and 5.5 Myr) are not directly
listed in Waltham (2015). The 11.5 Myr and 5.5 Myr cycles we identified are likely
comparable to the ~9 Myr and 4.8 Myr cycles widely documented in Phanerozoic
sedimentary records (e.g., Boulila et al., 2012; Ikeda et al., 2017; Ikeda and Tada,
2014; Laskar, 1990; Martinez and Dera, 2015), which are interpreted as modulation
periods of the 4.8 Myr and 2.4 Myr orbital eccentricity cycles, respectively. While
further detailed investigation would be required to fully confirm this
interpretation, we cite these previous studies to provide context. Our study
primarily focuses on the astronomical calibration of the Doushantuo Formation,
rather than the nature of these long-period modulations.

To address this point, we have revised the text read: "*The MTM power spectra*
*of the 405-kyr-tuned MS series in the three drillcores exhibit significant spectral*
*peaks at 405 kyr, 141-90 kyr, 32.9-28.3 kyr and 21.3-15.3 kyr (Supplementary Fig.*
*23), which align with major periods predicted for orbital forcing of early Ediacaran*
*solar insolation*⁴¹. *The spectra also reveal longer-period cycles of 11.5 Myr, 5.5 Myr,*
*and 2.4 Myr, which may correspond to the modulation periods of the ~9 Myr, ~4.5*
*Myr and ~2.4 Myr orbital eccentricity cycles documented in Phanerozoic sedimentary*
*records, respectively (e.g., refs. 43–45).*" See **Lines 204-207** in the main text.

**Comment 5:** Line 370: *The MS resolution of 2 cm implies that some precession cycles*
*have very few (<5) measurements per cycle. I don't think this hinders the*
*astrochronological interpretation, which is based on the 405-kyr cycle, but it should*
*perhaps be stated somewhere for transparency.*

**Reply:** Thank you for this thoughtful consideration regarding sampling resolution.
You raise a valid point about the potential limitations of our 2-cm sampling
resolution for detecting precession-scale cycles. In Member II, where
sedimentation rates are relatively low, this resolution may indeed result in
relatively few measurements per precession cycle. However, this does not hinder
our primary astronomical calibration, which is based on the well-resolved 405-kyr

eccentricity cycles. The longer periodicities (405-kyr and 100-kyr cycles) are
robustly represented in our dataset, with numerous measurements per cycle.

To address this point, we have added a clarifying statement in the "Samples
and Magnetic susceptibility measurement" section: "*While the 2-cm sampling
resolution may result in relatively few measurements for precession-scale cycles
within Member II (Supplementary Fig. 8) due to its relatively low sedimentation rates
(see the "Estimation of average sedimentation rate" section for details), this does not
affect our astronomical calibration which is primarily based on the well-resolved 405-
173 kyr eccentricity cycles.*" See **Lines 403-407** in the main text.

**Comment 6:** Line 425: *It might be good to add that the authors implicitly assume that
MS mostly reflects a fluvial, rather than an aeolian signal. I infer this from the usage
of decreases in MS (and $^{87}\text{Sr}/^{86}\text{Sr}$) as an indicator for the onset of the Gaskiers
glaciation (Supplementary materials lines 381-383). An aeolian MS signal would be
expected to increase during a glacial period – although this drying signal may not be
as significant in a world without land vegetation.*

**Reply:** Thank you for this insightful comment regarding the interpretation of our
MS signals. This is an important point about the potential contributions of fluvial
versus aeolian processes to MS variations, particularly during glacial periods.

In our study, we adopt the widely accepted interpretation that MS
predominantly reflects detrital fluxes from terrestrial sources transported via
fluvial processes in marine environments (e.g., Kodama and Hinnov, 2014). While
aeolian input could potentially increase during glacial periods, the consistent
decrease in multiple paleoclimatic proxies (MS and $^{87}\text{Sr}/^{86}\text{Sr}$) during the Gaskiers
interval and the sedimentological context of our studied sections strongly support
the interpretation of MS variations as primarily reflecting changes in fluvial input.

To address this point more explicitly, we have revised the relevant section to
state: "*Magnetic susceptibility (MS) measures the concentration of magnetic
minerals in sediments⁶⁵. It is widely used as a proxy for detrital fluxes from terrestrial
sources, primarily transported via fluvial processes in marine environments^{65,66}.*

*Aeolian inputs can also contribute to MS signals, particularly during glacial periods.*
*However, during the Gaskiers interval, multiple continental paleoclimate proxies*
*(e.g., MS, ⁸⁷Sr/⁸⁶Sr ratios, and the Chemical Index of Alteration; Supplementary Fig.*
*25) show notable decreases. Combined with the sedimentological context, these*
*observations strongly support a predominantly fluvial origin for the MS variations in*
*this study. " See **Lines 461-468** in the main text.*

**Response to the supplementary materials:**

***Comment 7:** Line 51: The name 'Shuram excursion' is not mentioned in this paragraph*
*as a probable synonym for EN3, but it is mentioned in Supplementary Figure 2. I would*
*add it in this paragraph for completeness; from my (limited) knowledge of Ediacaran*
*isotope stratigraphy, it is a commonly used term.*

**Reply:** Thank you for pointing out this omission and for highlighting the
importance of including the term "Shuram excursion." We agree that it is a widely
recognized and commonly used term in Ediacaran isotope stratigraphy and that
referencing it in the main text would improve the completeness and clarity of our
discussion.

To address this, we have revised the paragraph to explicitly mention the
Shuram excursion as a synonym for EN3. The updated text now reads:

" *The EN3 event, also widely referred to as the Shuram excursion, represents*
*the largest negative $\delta^{13}\text{C}$ excursion in Earth's history and is globally recognized in*
*carbonate successions¹⁴." See **Lines 54-56** in the Supplementary Information.*

***Comment 8:** Line 187-193: This phrasing makes it sound as if the presence of*
*astronomical cycles is assumed as given. So, once you know the sedimentation rates,*
*you can assign any frequency that fits to the right astronomical parameter. But we do*
*not know whether a certain sedimentary succession has preserved astronomical cycles*
*to such a degree that they can be extracted. I think there is plenty additional evidence*
*that these are indeed astronomical cycles, as is shown later in the manuscript: e.g., the*

405-kyr signal shows a clear 2.4 Myr amplitude modulation; MS is a proxy that can
and has been shown to record all these astronomical parameters in similar
environments; and also the sheer length of the record implies that it would be stranger
to NOT detect any astronomical signal whatsoever within this interval. It should just be
stated a bit more explicitly that the authors are not blindly assuming the presence of
astronomical signals (which I think they certainly do not).

**Reply:** Thanks for your thoughtful comment regarding the methodology of
astronomical cycle identification. It is crucial to clearly demonstrate, rather than
assume, the presence of astronomical signals in sedimentary records.

We agree that our original phrasing could be interpreted as assuming the
presence of astronomical cycles. To better reflect our rigorous approach to cycle
identification, we have revised this section to read:

*"To summarize, multiple lines of evidence support the astronomical origin of*
*the observed cycles: (1) the presence of characteristic frequency ratios matching*
*theoretical Milankovitch periods for the early Ediacaran Period (405 kyr, ~100 kyr,*
*~31.2 kyr, and ~18 kyr cycles)⁵¹, (2) the observation of expected amplitude*
*modulations (e.g., the ~2.4 and ~1.2 Myr modulation of the 405-kyr eccentricity and*
*~31.2-kyr obliquity cycle, respectively; Supplementary Fig. 23), (3) the demonstrated*
*sensitivity of MS to astronomical forcing in similar depositional environments^{55,56},*
*and (4) the extended length of our records (>8 Myr) which increases the statistical*
*reliability of detected cycles (Fig. 3)." See Lines 329-337 in the Supplementary*
**Information.**

**Comment 9:** *Supplementary Note 3 in general: Regarding sedimentation rate changes*
*(which I think are addressed well, cyclostratigraphy-wise), I miss a discussion on how*
*these relate to lithological changes and whether the sedimentation rate changes make*
*sense in that context.*

**Reply:** Thanks for your important suggestion regarding the relationship between
sedimentation rate changes and lithological variations. This is a valuable point

about the need to integrate sedimentological context with our cyclostratigraphic
interpretations.

To address this, we have expanded Supplementary Note 3.2 to include a
detailed discussion linking sedimentation rate variations to lithological changes
and sequence stratigraphic context. The revised text now reads: “*Our analysis*
*reveals systematic variations in sedimentation rates derived from the maxima of 405-*
*kyr cycles that integrate paleogeographic setting, lithological variation and sequence*
*stratigraphic framework. The average sedimentation rates decrease basinward from*
*WD1 (shallow intra-shelf basin, averaging ~0.4 cm/kyr) through EYC2 (deep intra-*
*shelf basin, averaging ~0.36 cm/kyr) to ZK68 (lower slope setting, averaging ~0.17*
*cm/kyr) (Supplementary Table 2 and Supplementary Fig. 24). These variations are*
*consistent with depositional gradients and paleogeographic positions. Sedimentation*
*rates also show systematic variations within depositional sequences. Rates peak*
*during early transgressive systems tracts (TST), where carbonate-rich units dominate,*
*reflecting higher accommodation, increased carbonate production, and enhanced*
*depositional energy (Supplementary Fig. 24). In contrast, during highstand systems*
*tracts (HST), sedimentation rates are generally lower, especially in mudstone-*
*dominated intervals, reflecting reduced terrigenous input and weakened*
*accommodation space under high sea-level conditions (Supplementary Fig. 24).*
*These lithological and sedimentation rate changes further reflect broader*
*paleoenvironmental shifts, such as sea-level fluctuations and variations in sediment*
*supply, which are consistent with the expected impacts of Milankovitch-scale climatic*
*cycles on depositional systems.” See **Lines 338-353** in the Supplementary*

**Information.**

To better illustrate these relationships, we have added a new figure
(Supplementary Fig. 24) comparing sedimentation rates derived from 405-kyr
cycle maxima with paleogeographic positions, lithofacies, and regional sequence
stratigraphic framework across the studied sections. See **Line 2855** in the
**Supplementary Information.**

**Comment 10:** *Line 2736 (Supplementary Fig. 23): For visual clarity, it would be nice*
*to mark the identified astronomical parameters on the eFFT plots here, similar to how*
*it was shown in Supplementary Fig. 9c.*

**Reply:** Thank you for this helpful suggestion to improve the visual clarity of
**Supplementary Figure 23. We have modified the eFFT plots to include labels for**
**the identified astronomical parameters, following the format used in**
**Supplementary Figure 9c. This addition will help readers more easily identify and**
**compare the key orbital frequencies across different analyses. See Line 2846 in the**
**Supplementary Information.**

**Response to specific comments highlighted in the main text PDF by Dr. Nina**
**Wichern (Reviewer 1).**

**Comment 11:** *Line 33 I propose to add ", based on key sections from South China,"*
*here or elsewhere early in the abstract for clarity.*

**Reply:** Thank you for this constructive suggestion. We have modified the abstract
**by adding "anchored by the radioisotopic date of the Gaskiers glaciation onset, based**
**on key sections from South China." See Lines 34-35 in the main text.**

**Comment 12:** *Line 233 "ages" duplicate*

**Reply:** Thank you for catching this error. We have removed the duplicate word
**"ages". See Line 265 in the main text.**

**Comment 13:** *Line 509 Is there a lot of text missing here?*

**Reply:** Thank you for pointing out this issue. Upon investigation, we found that a
**portion of text at Line 509 was inadvertently omitted during the manuscript**
**conversion from Word to PDF in the submission system.**

**The complete content should read:**

***"Age constraints for the $^{87}\text{Sr}/^{86}\text{Sr}$ ratios and $\delta^{13}\text{C}_{\text{carb}}$ curves are derived***
***assuming constant sedimentation rates between radioisotopic dates and***

*astrochronological tie points, as detailed in Supplementary Table 3 and*
*Supplementary Table 4. The same color is used to plot both ages and*
*chemostratigraphic data from the same location. OOE oceanic oxygenation event."*
**See Lines 640-644 in the main text.**

**We have thoroughly reviewed the entire manuscript, restored the missing text**
**in the revised version, and implemented additional measures to prevent any**
**content loss during future file conversions.**

**Response to specific comments highlighted in the Supplementary Information**
**PDF by Dr. Nina Wichern (Reviewer 1).**

**Comment 14:** *Line 381-383 "decreases in magnetic susceptibility and $^{87}\text{Sr}/^{86}\text{Sr}$ ratios,*
*reflecting the emergence of weakened continental weathering and reduced terrestrial*
*detrital flux, respectively." This order makes it sound as if MS tracks continental*
*weathering and Sr isotopes track terrestrial flux, when it is the other way around as*
*described in the main text.*

**Reply:** **Thank you for highlighting this important clarification. We have revised**
**the statement to ensure that the correct relationship is accurately conveyed. The**
**updated text now reads:**

*"decreases in magnetic susceptibility and $^{87}\text{Sr}/^{86}\text{Sr}$ ratios, reflecting reduced*
*terrestrial detrital flux and the emergence of weakened continental weathering,*
*respectively." See Lines 408-409 in the Supplementary Information.*

**We appreciate your careful attention to detail, which has helped improve the**
**clarity and accuracy of our manuscript.**

**Response to Reviewer 2**

**We sincerely thank Reviewer 2 for your thorough and constructive review of**
**our manuscript. We deeply appreciate your recognition of the potential**
**significance of our work in establishing a ~60-million-year astrochronology for the**
**early Ediacaran. At the same time, we are grateful for your insightful questions**
**regarding our methodology and interpretations, which have highlighted areas that**
**warrant further clarification and refinement. In particular, your detailed**
**comments on the cyclostratigraphic approach and the continuity of the succession**
**have been invaluable in helping us improve the clarity, rigor, and overall**
**robustness of our analysis.**

**Below, we provide a point-by-point response to each of your concerns, with a**
**special focus on the statistical methods employed in our study and the evidence**
**supporting stratigraphic continuity.**

***Comment 1:** Cyclostratigraphic analysis in SI. As stated before the important*
*details of the cyclostratigraphic analysis are “hidden” in the SI which has the risk that*
*readers will not pay that much attention to it, while these details are of critical*
*importance for the ms. But this is not for the authors to blame, but in my opinion a*
*general problem with many papers today. Now only a short summary is given in the ms*
*itself and a single review figure with the important results of bandpass filtering in the*
*depth and time domain. Unfortunately, this important figure was rather difficult to view*
*as it is somewhat blurred in the pdf.*

**Reply:** **We sincerely thank you for highlighting the importance of making the**
**cyclostratigraphic analysis more accessible and visible. We agree that presenting**
**these methodological details more prominently in the manuscript is crucial for**
**improving its clarity, accessibility, reproducibility, and scientific impact.**
**Additionally, we acknowledge the issue with the figure resolution and its**
**importance for readers to clearly interpret the results.**

**To address these concerns, we have implemented the following improvements:**

**(1) Expanded description in the "Time Series Methods" section of the Methods:**

*"Average sedimentation rates were derived through integrated analysis of*
*independent age constraints and TimeOpt statistical modeling. These rates, combined*
*with theoretical Milankovitch frequency ratios characteristic of the early Ediacaran*
*Period (detailed in Supplementary Note 2), enabled the attribution of identified*
*dominant frequencies to specific astronomical cycles. The astronomical signals were*
*extracted using Gaussian bandpass filtering⁶⁴. Astronomical tuning was performed*
*using the stable 405-kyr eccentricity cycle as a primary target, implemented through*
*the 'Age Scale' function in Acycle, facilitating the transformation from depth to time*
*domain and generating a floating astronomical time scale. This floating time scale*
*was subsequently calibrated against high-precision radioisotopic dates to establish*
*an absolute astronomical time scale."* See **Lines 482-491** in the main text.

*"TimeOpt is a statistical method for optimizing sedimentation rate estimation*
*through orbital signal analysis^{70,71}. This approach converts proxy series from depth*
*to time domain at various test sedimentation rates and employs Taner filtering and*
*Hilbert transforms to isolate potential precession cycles. The method generates two*
*correlation coefficients: $r^2_{envelope}$ (correlation between the amplitude envelope and*
*theoretical eccentricity) and r^2_{power} (correlation with combined eccentricity-*
*precession frequencies). The product of these coefficients ($r^2_{opt} = r^2_{envelope} \times$
r^2_{power}) determines the optimal sedimentation rate, with the highest r^2_{opt} value
indicating the most probable rate. Statistical significance is assessed through Monte
Carlo simulations using red noise series, with p -value <0.05 considered statistically
significant." See **Lines 492-500** in the main text.*

These additions provide a streamlined yet comprehensive overview of our
methodology, including sedimentation rate determination, identification of orbital
cycles, and construction of the time scale. This ensures that the critical details are
no longer "hidden" in the supplementary materials.

(2) Explanation of TimeOpt principles and statistical evaluation:

To further clarify the methodology, we have included a detailed explanation
of the theoretical principles underlying the TimeOpt method in the "Time Series
Methods" section. This explanation outlines how TimeOpt optimizes

sedimentation rates by testing a range of possible rates to align observed
stratigraphic cycles with known orbital periodicities.

We also describe the statistical evaluation of sedimentation rates using
parameters such as r^2_{opt} and p-values, which measure both the strength of the fit
and its significance. Key results from the TimeOpt analysis, previously confined to
the Supplementary Information, have now been integrated into the main text. This
includes the optimal sedimentation rates derived for each subset of the dataset and
their corresponding null hypothesis significance levels. These updates ensure that
readers have direct access to critical analytical details. See **Lines 504-523** in the
main text.

**(3) Integration of Supplementary Note 3.1 into the main text:**

To further enhance transparency, we have incorporated Supplementary Note
3.1 (“Estimation of average sedimentation rate” section) into the “Methods”
section in the main text. This addition provides a clear demonstration of how the
sedimentation rate for the studied interval was determined, allowing readers to
better evaluate the robustness of our cyclostratigraphic framework. See **Lines 524-**
**567** in the main text.

**(4) Optimized Figure presentation:**

We have improved the resolution and clarity of the key figure illustrating the
results of bandpass filtering in both depth and time domains. Specific
optimizations include:

(a) Enhanced image resolution for sharper visualization.

(b) Refined annotations to highlight key features of the plots.

(c) Adjustments to ensure optimal quality in the PDF output.

These improvements ensure that the figure is easier to interpret and visually
accessible across different platforms. See **Lines 583 and 614** in the main text.

*Comment 2: Very low sedimentation rates: This is not a point of critics on the*
*cyclostratigraphy but the authors have to clarify in the ms why the sedimentation rates*
*(of less than 1 cm/kyr) are so low and whether the succession is essentially continuous*

*on the Milankovitch scale. Is that normal? My experience is that sedimentation rates*
*are generally higher. The risk exists that the succession contains hidden unconformities*
*and hiatuses so that the average sedimentation rate becomes lower. This principle is*
*outlined among others in Andrew Miall papers (his Sedimentation Rate Scale (SRS)).*
*The authors have to provide arguments for the continuity of the studied succession.*
*However, most if not all of the Doushantuo Fm is deeper marine so will be much less*
*prone to erosion and might therefore well be continuous and also explain the low*
*sedimentation rate. But since it would be such an amazing succession, this has to be*
*clarified in the main text.*

**Reply:** Thank you for raising this critical point regarding the very low
sedimentation rates observed in our study and for emphasizing the importance of
clarifying the continuity of the succession. We also appreciate the reference to
Andrew Miall's Sedimentation Rate Scale (SRS), which provides a valuable
framework for assessing the potential risks of hidden unconformities and hiatuses
in sedimentary successions.

Below, we address your concerns and provide additional clarification
regarding the continuity of the succession, the geological context of low
sedimentation rates, and independent validation of these rates. We have also
incorporated these explanations into the main text.

**(1) Continuity of the succession:**

Previous sedimentological and sequence stratigraphic studies have
documented no evidence for stratigraphic gaps within Members I and II of the
Doushantuo Formation in the Yangtze platform's intrashelf basin and slope
environments (e.g., Jiang et al., 2011; Zhu et al., 2013). While Li et al.(2024) recently
suggested potential significant hiatuses near the boundary between Members II
and III- a concern also raised by Dr. Nina Wichern (Reviewer 1), detailed
examination of our study core reveals no sedimentological evidence for
stratigraphic gaps across the boundary between Members I and II (**Response-**
**Figure 1**). Besides, we provide an alternative interpretation of the data presented
in Li et al. (2024) based on our observations and understanding (see **Lines 84-128**

**in this Response** for details). Thus, based on the available sedimentological
evidence and previous studies, we hold the opinion that studied succession is
essentially continuous on the Milankovitch scale. We have added these points to
the manuscript for clarification. See **Line 134-136** in the main text.

**(2) Geological context of low sedimentation rates:**

The exceptionally low sedimentation rates observed in Member II (<1 cm/kyr)
can be explained by several geological and paleogeographic factors:

**(a) Post-Marinoan transgression and highstand systems tract:**

Following the deposition of the Member I cap dolostone sequence, which
marks the terminal phase of the Marinoan deglaciation, Member II accumulated
during a period of rapid global sea-level rise that established a highstand systems
tract (Jiang et al., 2011; McFadden et al., 2008). This transgressive episode
submerged paleohighs on the Yangtze Platform, reducing the contribution of
terrigenous sediments.

**(b) Paleogeographic position and sediment supply:**

During Member II deposition (ca. 635–580 Ma), the South China Block was
positioned at mid-latitudes (Zhao et al., 2018), which likely limited terrigenous
input due to reduced weathering rates. Additionally, the depositional setting
(intrashelf basin and slope environments) was characterized by low-energy
conditions, further contributing to reduced sediment supply.

**(c) Reduced carbonate productivity:**

Member II primarily consists of fine-grained siliciclastic and carbonate
sediments, deposited in distal, low-energy settings. Previous studies (e.g., Xiao et
al., 2021) have suggested that the low sedimentation rates in these settings may
reflect deposition within the outer zone of non-skeletal carbonate accumulation
(paleolatitude $\geq 30^\circ$), where reduced carbonate oversaturation limits carbonate
productivity.

**(3) Observed depositional gradients:**

The average sedimentation rates decrease basinward from WD1 (shallow
intra-shelf basin, averaging ~0.4 cm/kyr) through EYC2 (deep intra-shelf basin,

averaging ~ 0.36 cm/kyr) to ZK68 (lower slope setting, averaging ~ 0.17 cm/kyr)
(Supplementary Table 2 and Reply-Figure 3). These variations are consistent with
the paleogeographic gradient and depositional environment within the Yangtze
Platform. Thus, the observed sedimentation rates are not anomalous but instead
reflect the geological and depositional context of Member II. We have clarified
these points in the main text.

(3) Independent validation of sedimentation rates:

Our sedimentation rate estimates are supported by independent radiometric
dating studies:

At the Jiulongwan section, located in the deeper intrashelf basin on the
Yangtze Platform relative to the EYC2 drillcore (see Reply-Figure 3),
sedimentation rates in the lower part of Member II have been independently
estimated at ~ 0.23 cm/kyr based on radiometric dating (Sui et al., 2018). This is
consistent with the sedimentation rates observed in our cores WD1 (~ 0.4 cm/kyr)
and EYC2 (~ 0.36 cm/kyr). Thus, the observed sedimentation rates align with the
depositional gradient across intrashelf basin on the Yangtze Platform, further
supporting the validity of our estimates.

**Reply-Figure 3** Sedimentary facies variations of the Doushantuo Formation along
a transect from northwest to southeast across the central Yangtze Platform
(modified from Zhu et al. (2013)). JLW Jiulongwan.

(4) Addition to the Manuscript:

To clarify these points in the main text, we have added the following
explanation: *"The exceptionally low sedimentation rates during Member II*
*deposition can be attributed to multiple geological factors. Rapid sea-level rise*
*following the Marinoan deglaciation established a highstand systems tract on the*
*Yangtze Platform^{1,2,27}, submerging paleohighs and reducing terrigenous input due to*

*the South China Block's mid-latitude position (ca. 35–45°N)³⁶ during ca. 635-580*
*Ma. The intrashelf basin and slope environments were further characterized by*
*limited sediment supply and reduced carbonate productivity, consistent with*
*deposition in the outer zone of non-skeletal carbonate accumulation (paleolatitude*
*≥30°)⁴², where reduced carbonate oversaturation further contributed to low*
*sedimentation rates. This interpretation is corroborated by radiometric dating at the*
*Jiulongwan section, a key intrashelf basin site on the Yangtze Platform, where*
*sedimentation rates in the lower Member II are similarly low (~0.23 cm/kyr)²⁴.” See*
**Lines 185-195 in main text.**

**Comment 3:** *The authors split up the succession in distinct intervals with specific*
*sedimentation rates. However, this approach assumes abrupt changes in sedimentation*
*rate which is less likely. The selection of the intervals are based on ESA but this is a*
*windowed approach (they used quite long windows) that will not provide clear evidence*
*for such abrupt changes.*

**Reply:** **Thank you for this insightful comment regarding the potential assumption**
**of abrupt sedimentation rate changes and the limitations of using a windowed**
**approach in our analysis. We appreciate the opportunity to clarify our**
**methodology and address the concerns raised.**

**(1) Clarification of interval subdivisions**

**The subdivision of the succession into distinct intervals in our study was not**
**based solely on the results of Evolutive Fast Fourier Transform (eFFT) analysis,**
**which inherently smooths transitions and limits the resolution of finer-scale**
**variations. Instead, we used an integrated, multi-proxy approach to identify**
**meaningful subdivisions.**

**Specifically, the intervals were defined based on:**

**(a) Frequency variations detected by eFFT analysis:**

**The eFFT analysis revealed variations in dominant frequencies, which**
**provided evidence for potential sedimentation rate changes. These frequency shifts,**

while smoothed due to the use of relatively long sliding windows, were interpreted
as reflecting average trends in depositional conditions over stratigraphic intervals.

**(b) Distinct lithofacies changes observed in the drillcores:**

Lithofacies transitions in the cores were used to identify shifts in depositional
environments. These changes were interpreted as reflecting gradual variations in
sediment supply, energy conditions, or depositional settings, rather than abrupt
changes.

**(c) Recognized lithostratigraphic boundaries:**

The identified intervals were also aligned with well-documented
lithostratigraphic boundaries in the Doushantuo Formation, which have been
previously established through detailed sedimentological and stratigraphic studies.

By integrating these three lines of evidence, we ensured that the subdivisions
are geologically meaningful and reflect actual depositional changes rather than
artifacts of the spectral analysis. This comprehensive approach avoids the
assumption of abrupt changes in sedimentation rates and provides a robust
framework for cyclostratigraphic analysis.

While we acknowledge that the use of relatively long sliding windows in eFFT
is necessary to reliably capture dominant Milankovitch-scale periodicities,
particularly the 405-kyr eccentricity cycles, this approach inherently smooths
transitions and limits the resolution of finer-scale variations. As a result, the
sedimentation rate variations indicated by this method represent average values
across stratigraphic intervals, reflecting gradual changes in depositional
environments and sediment supply conditions. These interval values should be
interpreted as smoothed trends rather than abrupt shifts. To further validate the
sedimentation rate estimates for each interval, we independently applied TimeOpt
analysis, which provides additional support for the robustness of the subdivision
scheme and the sedimentation rates derived for each interval.

To address the reviewer's concerns and improve clarity, we have revised the
manuscript to explicitly describe this integrated methodology and emphasize the

**nature of the identified intervals. The following text has been added to the Methods**
**section:**

*"Evolutionary Fast Fourier Transform (eFFT) analysis revealed frequency*
*variations, which may reflect sedimentation rate changes. These variations, along*
*with observed lithofacies changes and lithostratigraphic boundaries, served as a*
*framework for cyclostratigraphic analyses of multiple subsets, including five from*
*WD1 and seven from ZK68 (see Supplementary Note 3 for details and Figs. 7b, 8b*
*and 9b)". See **Lines 167-171** in the main text.*

**Comment 4:** *TimeOpt. The authors subsequently use TimeOpt with constant*
*sedimentation rates to compute the optimal sedimentation rate for each of the*
*previously selected intervals. However, they use a very short range of possible*
*sedimentation rates that only covers the expected rate based on the eFFT analysis (?).*
*In my opinion, they should use a much wider range to allow other possible*
*sedimentation rates. This is especially important in view of point 1. However, their*
*argument might be that they already used that wider range already in the eFFT analysis,*
*but eFFT does not provide estimates of the most likely (optimal) sedimentation rate.*
*They base their interpretation on their own interpretation of the cycles ratios under*
*changing sedimentation rates.*

**Reply:** **Thank you for this valuable comment regarding the range of**
**sedimentation rates tested in our TimeOpt analysis. Below, we provide**
**clarification on our approach and address the specific concerns raised.**

**(1) Clarification of the approach**

**Our selection of sedimentation rate ranges for TimeOpt analysis was guided**
**by an integrated approach, combining constraints from preliminary eFFT analysis,**
**radiometric dating, geological context, and statistical significance criteria within**
**the TimeOpt framework. The sedimentation rates yielding the highest r^2_{opt} values**
**in the TimeOpt analysis were identified as the optimal sedimentation rates for each**
**interval. These rates were deemed robust and statistically significant, as they**

satisfied the criterion of P-value < 0.05 (rejecting the null hypothesis), providing
strong support for our sedimentation rate estimates.

We acknowledge your suggestion to use a broader range in the TimeOpt
analysis. Our selected ranges were chosen carefully with geological prior and
rigorous tests. Below, we outline the key considerations that informed our choice
of sedimentation rate ranges:

**(1) Preliminary constraints from eFFT analysis:**

eFFT analysis inherently considers a broad range of plausible sedimentation
rates by identifying frequency patterns linked to Milankovitch-scale
sedimentation rate variability. Although eFFT does not directly determine the
most likely sedimentation rates, it provides reasonable preliminary constraints,
which were further refined using TimeOpt analysis. These constraints allowed us
to narrow the tested ranges to focus on geologically reasonable sedimentation rates,
informed by the frequency ratios derived from eFFT.

**(2) Radiometric dating constraints:**

Radiometric dating at the base and top of the studied succession provided
independent temporal constraints, ensuring that the selected sedimentation rate
ranges were consistent with the overall depositional timeframe. These age controls
were critical for anchoring the cyclostratigraphic interpretations within a realistic
temporal framework and avoided testing sedimentation rates that were
inconsistent with the known age constraints.

**(3) Geological and regional context:**

The depositional setting of the Doushantuo Formation, particularly in the
Jiulongwan section and other intrashelf basin sites of the Yangtze Platform,
provided additional context for constraining plausible sedimentation rates.
Previous studies (e.g., Sui et al., 2018) have documented exceptionally low
sedimentation rates (~0.2–0.5 cm/kyr) in these settings, consistent with our
estimated ranges. These geological considerations ensured that the tested ranges
aligned with established knowledge of comparable depositional environments.

**(4) Statistical validation of sedimentation rates**

During our TimeOpt analysis, we tested multiple sedimentation rate ranges
for each interval. However, ranges that deviated significantly from those selected
produced suboptimal results, with lower r^2_{opt} values or higher P-values that failed
to meet the statistical significance threshold. Consequently, we selected the
sedimentation rate ranges that maximized r^2_{opt} values and yielded statistically
significant P-values. By contrast, the selected sedimentation rate ranges
consistently maximized r^2_{opt} values and produced statistically significant P-values,
confirming their validity.

(5) Additional analysis with expanded ranges

To further address the reviewer's concern, we conducted additional TimeOpt
analyses using **expanded sedimentation rate ranges** for two representative
intervals (subsets D2-4 and D3 from WD1 drillcore). These expanded ranges
covered sedimentation rates significantly higher and lower than those originally
tested. The results, presented in **Reply-Figure 4**, provide the following key insights:

(a) Consistency of optimal sedimentation rates:

Across the expanded ranges, the optimal sedimentation rates remained
consistent with our original estimates. This confirms the robustness of our
approach and the reliability of the selected ranges.

(b) Suboptimal results outside the selected range:

Sedimentation rates significantly higher or lower than the selected range
produced lower r^2_{opt} values and/or higher P-values that failed to meet the
statistical significance threshold. This outcome demonstrates that the broader
ranges tested did not yield better results and supports our original selection of
sedimentation rate ranges.

(c) Statistical validation:

The sedimentation rates identified within the originally selected range
consistently yielded the highest r^2_{opt} values and/or statistically significant P-
values, providing strong evidence that the selected range effectively captures the
most likely sedimentation rates.

**Reply-Figure 4. TimeOpt results for varying sedimentation rate ranges, focusing**
 **on subsets D2-4 and D3 from the WD1 drillcore.**

**Revisions to the Manuscript:**

**To address the reviewer’s concern and improve clarity, we have revised the**
 **manuscript to include an explicit explanation of the methodology used to select**
 **sedimentation rate ranges and the additional validation analyses conducted. The**
 **following text has been added to the “Time series methods” section:**

*“TimeOpt analysis was used to refine sedimentation rate estimates for each*
 *subset, using initial constraints derived from eFFT analysis, radiometric dating, and*
 *geological context. While eFFT analysis provides a broad framework for potential*
 *sedimentation rates, TimeOpt identifies the most likely rates by maximizing the*
 *correlation between observed stratigraphic cycles and theoretical Milankovitch*

*periodicities (r^2_{opt}). The sedimentation rate ranges tested in TimeOpt were chosen to*
*balance geological realism with computational efficiency, focusing on rates*
*consistent with radiometric age constraints and the depositional context of the*
*Doushantuo Formation (e.g., intrashelf basin sedimentation rates of ~0.2–0.5 cm/kyr;*
*ref. 24)”. See **Lines 504-511** in the main text.*

**Comment 5:** *eTimeOpt and eCOCO. So my question is why the authors chose for the*
*combination of eFFT and TimeOpt and did not select continuous evolutive techniques*
*such as eTimeOpt and/or eCOCO. Especially eCOCO, which like eTimeOpt and eFFT*
*start from variable sedimentation rates and not a constant rate like TimeOpt. Of course*
*they used eFFT as a first step to subdivide the record, but eCOCO will provide optimal*
*sedimentation rates per window, so in fact it combines the eFFT with TimeOpt. There*
*is also the option in one of the programs to trace for instance the expression of the long*
*eccentricity cycle to create an astrochronology and this avoids the necessity to use*
*selected intervals with constant sedimentation rates. This may provide an alternative*
*for the bandpass filtering of the cycle that is supposedly related to long eccentricity and*
*shown in the summary figure in the ms, as it allows for variable sedimentation rates*
*that constant sedimentation rates per fixed interval. The authors again have to show*
*why they selected their approach of combining eFFT with TimeOpt, and not use*
*eTimeOpt and/or eCOCO.*

**Reply:** **Thank you for raising this important point regarding the selection of**
**methods in our cyclostratigraphic analysis and for suggesting alternative**
**approaches, such as eTimeOpt and eCOCO. Below, we provide clarification and**
**justification for our chosen methodology combining eFFT and TimeOpt, as well**
**as a discussion of the potential limitations of continuous evolutive techniques.**

**Why we chose eFFT and TimeOpt**

**Our decision to use eFFT and TimeOpt was based on a combination of**
**dataset-specific considerations, research objectives, and a critical evaluation of**
**available methods. The following points outline the rationale behind our choice:**

**(1) Data characteristics and geological context:**

The studied succession contains distinct lithological changes and
stratigraphic boundaries, suggesting relatively stable depositional conditions
within specific intervals. Using eFFT as an initial step allowed us to identify
dominant frequency patterns and assess their variability across the stratigraphic
succession. This provided a framework for subdividing the record into geologically
meaningful intervals with relatively constant sedimentation rates.

TimeOpt was then applied to refine sedimentation rates within these intervals
by optimizing the fit between observed stratigraphic cycles and known orbital
periodicities. This two-step approach ensured that our results were robust and
tailored to the characteristics of the dataset.

(2) Focus on long-term orbital cyclicity:

Our study focuses on long-term orbital cycles, particularly the 405-kyr
eccentricity cycle, as the tuning target. The assumption of constant sedimentation
rates within predefined intervals is appropriate for capturing long-term trends, as
the 405-kyr cycle is less sensitive to short-term variations in sedimentation rates.

While eCOCO and eTimeOpt allow for variable sedimentation rates across
sliding windows, we found that the constant rate assumption within geologically
defined intervals provided a simpler yet effective framework for achieving our
scientific objectives.

Besides, in the simulation of the most likely sedimentation rate estimates, we
attempted to use the eTimeOpt method to model the variation of sedimentation
rates with depth; however, the results did not appear to be ideal. To address this
issue, we discussed it with Prof. Stephen R. Meyers (the developer of the TimeOpt
method, University of Wisconsin-Madison, USA). He pointed out that this could
be related to the low signal-to-noise ratio of the climate proxy data, particularly
given that the data were derived from Ediacaran strata and the low sedimentation
rates further exacerbated the uncertainties. Nevertheless, if the sedimentation
rates estimated by the TimeOpt method exhibit a high r^2_{opt} and significant P
values, and are consistent with independent age data and geological context, the
results from TimeOpt can largely be considered reliable.

**(3) Limitations of COCO and eCOCO**

We acknowledge the advantages of continuous evolutive techniques such as
eCOCO, which allow for variable sedimentation rates without predefined
intervals. However, our preliminary testing of these methods revealed several
theoretical and practical limitations, which ultimately made them unsuitable for
application in this study.

**(a) Statistical significance testing issues with COCO and eCOCO:**

During preliminary analyses, we tested multiple methods, including COCO
and eCOCO (**Reply-Figures 5-15**), to cross-validate sedimentation rates and
evaluate the astronomical forcing hypothesis. The sedimentation rates derived
from COCO and eCOCO were generally consistent with those from TimeOpt
(**Reply-Figures 5-15**). However, both eCOCO and COCO suffer from statistical
significance testing challenges, particularly with regard to multiple hypothesis
testing across a range of sedimentation rates.

COCO and eCOCO fail to properly account for multiple hypothesis testing
across a range of sedimentation rates. This results in an inflated false positive rate,
where spurious astronomical signals are detected in purely stochastic datasets. For
example, an analysis of a random time series (AR1 test) using COCO incorrectly
identified a statistically significant signal (**Reply-Figure 16**). This issue arises
because COCO does not apply corrections for multiple testing, such as the
Bonferroni correction.

**(b) Theoretical flaws in target spectrum fitting:**

COCO and eCOCO rely on fitting the power spectrum of the dataset to a
predefined "target power spectrum" based on expected orbital periodicities
(**Reply-Figure 17**). However, this approach has two key flaws:

The relative power of orbital cycles (e.g., eccentricity, obliquity, and
precession) in the target spectrum must be predefined, but their actual relative
power in geological records is often unknown.

The fixed distribution of power in the target spectrum makes the method
 inflexible and prone to mismatches with real-world data, particularly in settings
 with noise or weaker astronomical signals.

(c) Theoretical and practical limitations of the correlation approach:

COCO and eCOCO use correlation coefficients in the frequency domain to
 assess the fit between observed and target spectra. However, correlation
 coefficients are highly sensitive to noise and variations in the relative power of
 spectral peaks, making the results unreliable in many cases.

These limitations led us to select the more robust and statistically reliable
 combination of eFFT and TimeOpt for this study.

**Reply-Figure 5** COCO and eCOCO sedimentation rate maps for the subset D2-1 of the WD1
 **drillcore.** a–c correlation coefficient, the null hypothesis (H_0) significance level and number of
 contributing astronomical parameters. d–f Evolutionary correlation coefficient, H_0 significance level
 and the number of contributing astronomical parameters shown with sedimentation rates picked
 from the maxima of 405-kyr cycles (black line). Both COCO and eCOCO analyses were performed

with 2000 Monto Carlo simulations, testing sedimentation rates ranging from 0 to 1.5 cm/kyr with
a step of 0.01 cm/ky. The eCOCO analysis was conducted with a 2.2 m sliding window, and 0.01 m
steps.

**Reply-Figure 6 COCO and eCOCO sedimentation rate maps for the subset D2-2 of the WD1**

**drillcore. a–c** correlation coefficient, the null hypothesis (H_0) significance level and number of

contributing astronomical parameters. **d–f** Evolutionary correlation coefficient, H_0 significance level

and the number of contributing astronomical parameters shown with sedimentation rates picked

from the maxima of 405-kyr cycles (black line). Both COCO and eCOCO analyses were performed

with 2000 Monto Carlo simulations, testing sedimentation rates ranging from 0 to 1.5 cm/kyr with

a step of 0.01 cm/ky. The eCOCO analysis was conducted with a 2.2 m sliding window, and 0.01 m

steps.

**Reply-Figure 7 COCO and eCOCO sedimentation rate maps for the subset D2-3 of the WD1**

**drillcore. a–c** correlation coefficient, the null hypothesis (H_0) significance level and number of

contributing astronomical parameters. **d–f** Evolutionary correlation coefficient, H_0 significance level

and the number of contributing astronomical parameters shown with sedimentation rates picked

from the maxima of 405-kyr cycles (black line). Both COCO and eCOCO analyses were performed

with 2000 Monto Carlo simulations, testing sedimentation rates ranging from 0 to 1.5 cm/kyr with

a step of 0.01 cm/ky. The eCOCO analysis was conducted with a 2.4 m sliding window, and 0.01 m

steps.

**Reply-Figure 8 COCO and eCOCO sedimentation rate maps for the subset D2-4 of the WD1**

**drillcore. a–c** correlation coefficient, the null hypothesis (H_0) significance level and number of

contributing astronomical parameters. **d–f** Evolutionary correlation coefficient, H_0 significance level

and the number of contributing astronomical parameters shown with sedimentation rates picked

from the maxima of 405-kyr cycles (black line). Both COCO and eCOCO analyses were performed

with 2000 Monte Carlo simulations, testing sedimentation rates ranging from 0 to 1.5 cm/kyr with

a step of 0.01 cm/ky. The eCOCO analysis was conducted with a 2.4 m sliding window, and 0.01 m

steps.

**Reply-Figure 9 COCO and eCOCO sedimentation rate maps for the subset D1 of the ZK68**835 **drillcore. a–c** correlation coefficient, the null hypothesis (H_0) significance level and number of836 contributing astronomical parameters. **d–f** Evolutionary correlation coefficient, H_0 significance level

and the number of contributing astronomical parameters shown with sedimentation rates picked

from the maxima of 405-kyr cycles (black line). Both COCO and eCOCO analyses were performed

with 2000 Monto Carlo simulations, testing sedimentation rates ranging from 0 to 1.5 cm/kyr with

a step of 0.01 cm/ky. The eCOCO analysis was conducted with a 1.5 m sliding window, and 0.01 m

steps.

**Reply-Figure 10 COCO and eCOCO sedimentation rate maps for the subset D2-1 of the**

**ZK68 drillcore. a–c** correlation coefficient, the null hypothesis (H_0) significance level and number

of contributing astronomical parameters. **d–f** Evolutionary correlation coefficient, H_0 significance

level and the number of contributing astronomical parameters shown with sedimentation rates

picked from the maxima of 405-kyr cycles (black line). Both COCO and eCOCO analyses were

performed with 2000 Monto Carlo simulations, testing sedimentation rates ranging from 0 to 1

850 cm/kyr with a step of 0.01 cm/ky. The eCOCO analysis was conducted with a 1.2 m sliding window,

and 0.01 m steps.

**Reply-Figure 11 COCO and eCOCO sedimentation rate maps for the subset D2-2 of the**

**ZK68 drillcore. a–c** correlation coefficient, the null hypothesis (H_0) significance level and number

of contributing astronomical parameters. **d–f** Evolutionary correlation coefficient, H_0 significance

level and the number of contributing astronomical parameters shown with sedimentation rates

picked from the maxima of 405-kyr cycles (black line). Both COCO and eCOCO analyses were

performed with 2000 Monto Carlo simulations, testing sedimentation rates ranging from 0 to 1

859 cm/kyr with a step of 0.01 cm/ky. The eCOCO analysis was conducted with a 1.2 m sliding window,

and 0.01 m steps.

**Reply-Figure 12 COCO and eCOCO sedimentation rate maps for the subset D2-3 of the**

**ZK68 drillcore. a–c** correlation coefficient, the null hypothesis (H_0) significance level and number

of contributing astronomical parameters. **d–f** Evolutionary correlation coefficient, H_0 significance

level and the number of contributing astronomical parameters shown with sedimentation rates

picked from the maxima of 405-kyr cycles (black line). Both COCO and eCOCO analyses were

performed with 2000 Monte Carlo simulations, testing sedimentation rates ranging from 0 to 1

876 cm/kyr with a step of 0.01 cm/ky. The eCOCO analysis was conducted with a 1.0 m sliding window,

and 0.01 m steps.

**Reply-Figure 13 COCO and eCOCO sedimentation rate maps for the subset D2-4 of the**

**ZK68 drillcore. a–c** correlation coefficient, the null hypothesis (H_0) significance level and number

of contributing astronomical parameters. **d–f** Evolutionary correlation coefficient, H_0 significance

level and the number of contributing astronomical parameters shown with sedimentation rates

picked from the maxima of 405-kyr cycles (black line). Both COCO and eCOCO analyses were

performed with 2000 Monto Carlo simulations, testing sedimentation rates ranging from 0 to 1

886 cm/kyr with a step of 0.01 cm/ky. The eCOCO analysis was conducted with a 1.0 m sliding window,

and 0.01 m steps.

**Reply-Figure 14 COCO and eCOCO sedimentation rate maps for the subset D2-5 of the**
 **ZK68 drillcore. a–c** correlation coefficient, the null hypothesis (H_0) significance level and number
 of contributing astronomical parameters. **d–f** Evolutionary correlation coefficient, H_0 significance
 level and the number of contributing astronomical parameters shown with sedimentation rates
 picked from the maxima of 405-kyr cycles (black line). Both COCO and eCOCO analyses were
 performed with 2000 Monto Carlo simulations, testing sedimentation rates ranging from 0 to 1
 896 cm/kyr with a step of 0.01 cm/ky. The eCOCO analysis was conducted with a 1.4 m sliding window,
 and 0.01 m steps.

Reply-Figure 15 COCO and eCOCO sedimentation rate maps for the EYC2 drillcore. a–c

correlation coefficient, the null hypothesis (H_0) significance level, and the number of contributing

astronomical parameters. **d–f** Evolutionary correlation coefficient, H_0 significance level and the

number of contributing astronomical parameters shown with sedimentation rates picked from the

maxima of 405-kyr cycles (black line). Both COCO and eCOCO analyses were performed with

2000 Monto Carlo simulations, testing sedimentation rates ranging from 0 to 1 cm/kyr with a step

of 0.01 cm/ky. The eCOCO analysis was conducted with a 2.0 m sliding window, and 0.01 m steps.

Reply-Figure 16. Acycle’s COCO analysis of the AR1test stochastic times series

Reply-Figure 17. Acycle’s plot of the ‘target power spectrum’ for COCO analysis (top plot), and the power spectrum of the AR1test stochastic times series (lower plot)

**Response to Prof. Kenneth P. Kodama (Reviewer 3)**

**We sincerely thank Prof. Kenneth P. Kodama for positive assessment of our**
**manuscript and recognition of its significance in establishing a high-resolution**
**chronostratigraphy for the early Ediacaran. We are encouraged by your**
**acknowledgment of the broad implications of our work, including constraining the**
**timing of the Gaskiers glaciation, Marinoan cap carbonate deposition, and early**
**multicellular life evolution.**

**Your validation of our approach, combining radiogenic dating with**
**statistically significant Milankovitch cycle identification, further reinforces the**
**reliability of our astronomically-calibrated chronology. We greatly appreciate**
**your thoughtful and supportive feedback, which motivates us to further pursue**
**this line of research.**

*Comment 1: The only scientific comment I have is not a criticism and should not hold*
*up publication of this important work. The sediment accumulation rates reported are*
*all less than 1 cm/kyr which from my experience is somewhat low. I have seen low rates*
*off the US east coast in the Miocene, so rates this low are not unheard of, but given that*
*these sediments were deposited in the Ediacaran when there was no terrestrial*
*vegetation to slow down erosion of the continents it is surprising. Maybe some comment*
*about this finding would be useful for the reader, but certainly not necessary.*

**Reply: Thank you for his thoughtful observation regarding the sediment**
**accumulation rates reported in our study. We agree that the rates, being less than**
**1 cm/kyr, are relatively low compared to some other sedimentary settings.**
**However, as you noted, low accumulation rates such as these are not**
**unprecedented, with examples from the Miocene off the US east coast providing a**
**useful comparison.**

**The relatively low sedimentation rates observed in our study likely reflect the**
**specific depositional and environmental context of South China during the early**
**Ediacaran. Several factors may contribute to these low rates:**

(1) Following the deposition of the Member I cap dolostone sequence, which
marks the terminal phase of the Marinoan deglaciation, Member II accumulated
during a period of rapid global sea-level rise that established a highstand systems
tract (Jiang et al., 2011; McFadden et al., 2008). During this transgressive episode,
the Yangtze Platform exhibited a northwest-to-southeast paleotopographic
gradient, submerging numerous paleohighs. The reduction in exposed source
areas, combined with the paleogeographic position of the South China Block at
mid-latitudes (ca. 35–45°N) during Member II deposition (Zhao et al., 2018), likely
resulted in diminished terrigenous input. Member II of the Doushantuo Formation
primarily consists of fine-grained siliciclastic and carbonate sediments, deposited
at exceptionally low sedimentation rates within intrashelf basin to distal slope
settings. These low rates are attributed to the limited terrigenous supply, reduced
carbonate productivity, and the low-energy conditions characteristic of these
environments. Furthermore, previous studies suggest that the unusually low
average sedimentation rates may reflect deposition within the outer zone of non-
skeletal carbonate accumulation (paleolatitude $\geq 30^\circ$), where reduced carbonate
oversaturation is expected (Xiao et al., 2021). The average sedimentation rates
decrease basinward from WD1 (shallow intra-shelf basin, averaging ~ 0.4 cm/kyr)
through EYC2 (deep intra-shelf basin, averaging ~ 0.36 cm/kyr) to ZK68 (lower
slope setting, averaging ~ 0.17 cm/kyr) (Supplementary Table 2 and
Supplementary Fig. 24). These variations are consistent with depositional
gradients and paleogeographic positions. Thus, such exceptionally low
sedimentation rates are consistent with both the geological and paleoclimatic
context.

(2) Additionally, at the Jiulongwan section, a representative section of the
Doushantuo Formation located in the deep intrashelf basin of the Yangtze
Platform, the sedimentation rate in the lower part of Member II has been
estimated at 0.23 cm/kyr based on radiometric dating (Sui et al., 2018). This rate
is comparable to the sedimentation rates observed in our cores WD1 and EYC2.

We have added the following text in the manuscript to clarify the potential
causes of such low sedimentation rates: " *The exceptionally low sedimentation rates*
*during Member II deposition can be attributed to multiple geological factors. Rapid*
*sea-level rise following the Marinoan deglaciation established a highstand systems*
*tract on the Yangtze Platform^{1,2,27}, submerging paleohighs and reducing terrigenous*
*input due to the South China Block's mid-latitude position (ca. 35–45°N)³⁶ during ca.*
*635-580 Ma. The intrashelf basin and slope environments were further characterized*
*by limited sediment supply and reduced carbonate productivity, consistent with*
*deposition in the outer zone of non-skeletal carbonate accumulation (paleolatitude*
*≥30°), where likely reduced carbonate oversaturation further contributed to low*
*sedimentation rates⁴². This interpretation is corroborated by radiometric dating at the*
*Jiulongwan section, a key intrashelf basin site on the Yangtze Platform, where*
*sedimentation rates in the lower Member II are similarly low (~0.23 cm/kyr)²⁴."* See
**Lines 185-195 in the main text.**

**References**

- Boulila, S., Galbrun, B., Laskar, J., Pälike, H., 2012. A~ 9 myr cycle in Cenozoic
$\delta^{13}\text{C}$ record and long-term orbital eccentricity modulation: Is there a link? Earth
Planet. Sci. Lett. 317, 273–281.
- Ikeda, M., Tada, R., 2014. A 70 million year astronomical time scale for the deep-sea
bedded chert sequence (Inuyama, Japan): Implications for Triassic–Jurassic
geochronology. Earth Planet. Sci. Lett. 399, 30–43.
- Ikeda, M., Tada, R., Ozaki, K., 2017. Astronomical pacing of the global silica cycle
recorded in Mesozoic bedded cherts. Nat. Commun. 8, 15532.
- Jiang, G., Shi, X., Zhang, S., Wang, Y., Xiao, S., 2011. Stratigraphy and
paleogeography of the Ediacaran Doushantuo Formation (ca. 635-551Ma) in
South China. Gondwana Res. 19, 831–849.
<https://doi.org/10.1016/j.gr.2011.01.006>

Kodama, K.P., Hinnov, L.A., 2014. Rock Magnetic Cyclostratigraphy, Rock
Magnetic Cyclostratigraphy. John Wiley & Sons.
<https://doi.org/10.1002/9781118561294>

Laskar, J., 1990. The chaotic motion of the solar system: A numerical estimate of the
size of the chaotic zones. *Icarus* 88, 266–291. [https://doi.org/10.1016/0019-](https://doi.org/10.1016/0019-1035(90)90084-M)
[1035\(90\)90084-M](https://doi.org/10.1016/0019-1035(90)90084-M)

Li, R., Zhou, X., Eddy, M.P., Ickert, R.B., Wang, Z., Tang, D., Huang, K.J., Peng, P.,
2024. Stratigraphic evidence for a major unconformity within the Ediacaran
System. *Earth Planet. Sci. Lett.* 636, 118715.
<https://doi.org/10.1016/j.epsl.2024.118715>

Martinez, M., Dera, G., 2015. Orbital pacing of carbon fluxes by a ~9-My eccentricity
cycle during the Mesozoic. *Proc. Natl. Acad. Sci. U. S. A.* 112, 12604–12609.
<https://doi.org/10.1073/pnas.1419946112>

McFadden, K.A., Huang, J., Chu, X., Jiang, G., Kaufman, A.J., Zhou, C., Yuan, X.,
Xiao, S., 2008. Pulsed oxidation and biological evolution in the Ediacaran
Doushantuo Formation. *Proc. Natl. Acad. Sci. U. S. A.* 105, 3197–3202.
<https://doi.org/10.1073/pnas.0708336105>

Sui, Y., Huang, C., Zhang, R., Wang, Z., Ogg, J., Kemp, D.B., 2018. Astronomical
time scale for the lower Doushantuo Formation of early Ediacaran, South China.
*Sci. Bull.* 63, 1485–1494. <https://doi.org/10.1016/j.scib.2018.10.010>

Xiao, D., Cao, J., Luo, B., Tan, X., Xiao, W., He, Y., Li, K., 2021. Neoproterozoic
postglacial paleoenvironment and hydrocarbon potential: A review and new
insights from the Doushantuo Formation Sichuan Basin, China. *Earth-Science*
*Rev.* 212, 103453. <https://doi.org/10.1016/j.earscirev.2020.103453>

Zhao, G., Wang, Y., Huang, B., Dong, Y., Li, S., Zhang, G., Yu, S., 2018. Geological
reconstructions of the East Asian blocks: From the breakup of Rodinia to the
assembly of Pangea. *Earth-Science Rev.* 186, 262–286.
<https://doi.org/10.1016/j.earscirev.2018.10.003>

Zhu, M., Lu, M., Zhang, J., Zhao, F., Li, G., Aihua, Y., Zhao, X., Zhao, M., 2013.
Carbon isotope chemostratigraphy and sedimentary facies evolution of the

Ediacaran Doushantuo Formation in western Hubei, South China. Precambrian
Res. 225, 7–28.

**Dear Reviewer,**

We sincerely thank the reviewer for their thoughtful and constructive
evaluation of our manuscript. We are pleased that our responses to the critical
points raised in the previous round were found convincing and that the
manuscript is now deemed suitable for publication following the resolution of
the remaining minor comments.

Your insightful suggestions have been invaluable in refining our work, and
we have carefully considered and addressed each of the new comments. Below,
we provide a detailed, point-by-point response, outlining the steps we have
taken to enhance the clarity, rigor, and transparency of the manuscript. To
facilitate the review process, all modifications are highlighted **in red text** for
easy reference. Line numbers mentioned in our responses correspond to **the**
**tracked-changes version** of the manuscript and supplementary materials.

We hope our revisions meet your expectations and further improve the
impact of our work. Thank you again for your careful evaluation and
constructive feedback. We look forward to your further consideration of our
manuscript.

Best regards,

Chao Ma

Professor

State Key Laboratory of Oil and Gas Reservoir Geology and Exploitation &
Institute of Sedimentary Geology, Chengdu University of Technology

No.1 East 3rd Road, Erxianqiao, Chenghua District, Chengdu 610059, China

Email: machao@cdut.edu.cn

**Comment 1:** *Low sed rates: Still the low sedimentation rates are exceptional*
*and I advise the authors to be very careful in their wording. This also holds for*
*their argument that radiometric dates provide evidence for this low*
*sedimentation rate because what do these tell us about potential hiatuses or*
*condensed intervals that are present in the succession? In addition the*
*detection of hiatuses is not always straightforward, also or especially not in*
*cores.*

**Reply:** We thank the reviewer for raising this important point regarding
the interpretation of low sedimentation rates and the potential for
hiatuses or condensed intervals in the succession. We agree that caution
is warranted in the phrasing of our conclusions and acknowledge the
complexities involved in identifying hiatuses, particularly in core records.

In response to this comment, we have carefully reviewed and refined the
relevant sections of the manuscript to ensure that our discussion remains
balanced and avoids overinterpretation. Specifically, we have added the
following statement to address this concern: “*Radiometric dating at the*
*Jiulongwan section, a key intrashelf basin site on the Yangtze Platform,*
*confirms similarly low sedimentation rates in the lower Member II (~0.23*
*cm/kyr)²⁴. Average sedimentation rates decrease basinward, from ~0.4*
*cm/kyr at WD1 (shallow intrashelf basin) to ~0.34 cm/kyr at EYC2 (deep*
*intrashelf basin) and ~0.17 cm/kyr at ZK68 (lower slope) (Supplementary*
*Table 2), aligning with the depositional environments and*
*paleogeography of the Yangtze Platform. Although detailed core*
*examinations revealed no evidence of small-scale stratigraphic hiatuses*
*or condensed intervals, detecting such features in core records remains*
*inherently challenging⁴⁵. Despite these uncertainties, the sedimentation*
*rates align with independent radiometric dating and support a robust*
*interpretation consistent with the geological context.” See Lines 199-208
in the main text.*

**Comment 2:** *Wider range of sed rates: The authors did a good job in providing*
*examples of running TimeOpt over a wider range of sedimentation rates. It*
*might be a good idea to include these in the SI, although I leave that to the*
*discretion of the authors. It is clear that the authors experimented with different*
*statistical techniques before developing and choosing their preferred approach.*

**Reply:** We thank you for positive feedback on our exploration of a wider
range of sedimentation rates using TimeOpt and for the suggestion to
include these additional examples in the Supplementary Information (SI).
We appreciate that our efforts to test various statistical techniques before
selecting the final approach were evident and valued.

To address this, we have added the following statement to the manuscript:

*“Besides, we conducted supplementary TimeOpt analyses using*
*expanded sedimentation rate ranges for two representative intervals*
*(subsets D2-4 and D3 from the WD1 drillcore), allowing for rates*
*significantly higher and lower than those initially tested. The results*
*showed that the optimal sedimentation rates remained consistent with*
*our original estimates, underscoring the robustness of our approach.*
*Sedimentation rates outside the originally selected range—either*
*significantly higher or lower—produced lower r^2_{opt} values and/or P -*
*values that did not meet statistical significance. This outcome supports*
*the validity of our original sedimentation rate selection, as rates within*
*the initially selected range consistently produced the highest r^2_{opt} values*
*and statistically significant P -values, capturing the most plausible*
*sedimentation rates”*. See Lines 578-587 in the main text.

Additionally, we have included the Supplementary TimeOpt results
for these expanded sedimentation rate ranges in the Supplementary
Information to enhance transparency and reproducibility. See Lines 3065-
3113 in the Supplementary Information (Supplementary Figs. 23 and 24).

We thank you again for this constructive suggestion, which has
helped to strengthen the manuscript and further demonstrate the
robustness of our approach.

**Comment 3:** (e)COCO: I agree with the authors that the use of eCOCO may
not be recommended as long as the theoretical shortcomings are not repaired.
It is very good to see that they contacted Steve Meyers on such critical issues.
However, despite these shortcomings, the somewhat confusing eCOCO results
likely also tells something about the overall quality of their data and how
straightforward - or not – these data can be interpreted in terms of Milankovitch
forcing. Moreover, one of the problems with cyclostratigraphy is that often more
cycle period ratios will closely fit the ratios of the astronomical cycles, resulting
in more than one possible outcome / interpretation. For this reason, an
integrated stratigraphic approach in particular using tools like radio-isotopic
dating is critical and it is this approach that the authors largely follow (but see
above).

**Reply:** We thank you for the insightful comments regarding the use of
eCOCO and its implications for interpreting Milankovitch forcing. While
we agree that eCOCO's theoretical shortcomings limit its reliability, we
also recognize that the somewhat confusing results may reflect the
complexity and preservation of the dataset, which can influence how
straightforwardly Milankovitch forcing can be interpreted.

To address this concern, we have added an amplitude modulation
analysis of short-wavelength signals identified in the depth domain to the
Methods section. Specifically, we applied the Hilbert transform approach
described by Grippo et al. (2013) to extract the enveloping curve of
bandpass-filtered signals. An MTM spectral analysis was then performed
on this envelope to identify the cycles of amplitude modulation. Since
eccentricity affects climate via its amplitude modulation of precession,
this method provides a diagnostic tool to evaluate such spectral behavior.

The following description has been added to the Methods section:

*"To further evaluate the reliability of our cyclostratigraphic*
*interpretation, the amplitude modulation of a bandpass signal was*
*analyzed by applying the Hilbert transform to extract the signal's*
*envelope in the depth domain⁷⁵. MTM spectral analysis of the envelope*
*was then conducted to identify the amplitude modulation cycles of the*
*shorter-wavelength signal. Given that eccentricity modulates precession*
*to drive climate change, the Hilbert transform offers a robust method to*
*test for such diagnostic spectral patterns^{40,76}". See Lines 531-536 in the*
**main text.**

The results demonstrate that the envelope of the putative precession
signal (~18.5 kyr), extracted using the Hilbert transform, is systematically
amplitude-modulated in bundles of ~5–6 cycles, consistent with
precession modulated by eccentricity (Supplementary Figs. 25-26).
Similarly, the envelope of the putative short eccentricity signal (~112 kyr)
is amplitude-modulated in bundles of ~3–4 cycles (Supplementary Figs.
25-26), consistent with short eccentricity (95.2, 99.6, 124.4, and 132.2 kyr)
modulated by long eccentricity (~405 kyr). Spectral analysis of the ~18.5
140 kyr and ~112 kyr Hilbert transform envelopes shows power in the short
and long eccentricity bands (Supplementary Figs. 25 and 27), confirming
that both short and long eccentricity signals can be extracted from
precession and short eccentricity. Thus, the magnetic susceptibility data
from Doushantuo Formation of WD1, EYC2 and ZK68 drillcores, South
China pass one of the most diagnostic tests for orbital forcing: short
eccentricity and long eccentricity extracted from precession and short
eccentricity. Due to its effect on Earth's equinoxes, eccentricity primarily
affects insolation due to its amplitude modulation of precession (Mitchell
et al., 2021, 2008). This phenomenon forms a cornerstone of Phanerozoic

**cyclostratigraphy, where eccentricity signals are typically extracted from**
**precession (Hinnov, 2013). See Lines 211-222 in the main text.**

**Besides, variations in magnetic susceptibility (MS) reveal lamination**
**cycles, providing important sedimentological evidence for Milankovitch-**
**driven processes recorded in the strata (see Fig. 1 in the main text). The**
**long eccentricity (E) and short eccentricity (e) cycles extracted from the**
**WD1 and ZK68 drillcores further support this conclusion, with specific**
**frequency bands identified using a Gaussian filter (See Supplementary**
**Figs. 25-27).**

**We thank you once again for raising this important point, as it allowed**
**us to further strengthen the robustness of our analysis and interpretation.**

**References**

- Hinnov, L.A., 2013. Cyclostratigraphy and its revolutionizing applications in
the earth and planetary sciences. *Bull. Geol. Soc. Am.* 125, 1703–1734.
<https://doi.org/10.1130/B30934.1>
- Mitchell, R.N., Bice, D.M., Montanari, A., Cleaveland, L.C., Christianson, K.T.,
Coccioni, R., Hinnov, L.A., 2008. Oceanic anoxic cycles? Orbital prelude
to the Bonarelli Level (OAE 2). *Earth Planet. Sci. Lett.* 267, 1–16.
<https://doi.org/10.1016/j.epsl.2007.11.026>
- Mitchell, R.N., Gernon, T.M., Cox, G.M., Nordsvan, A.R., Kirscher, U., Xuan,
C., Liu, Y., Liu, X., He, X., 2021. Orbital forcing of ice sheets during
snowball Earth. *Nat. Commun.* 12, 4187. [https://doi.org/10.1038/s41467-](https://doi.org/10.1038/s41467-021-24439-4)
[021-24439-4](https://doi.org/10.1038/s41467-021-24439-4)
